# Anti-resonance in developmental signaling regulates cell fate decisions

Samuel J Rosen[1†], Olivier Witteveen[2†], Naomi Baxter[3], Ryan S Lach[4], Erik Hopkins[3], Marianne Bauer[2]*, Maxwell Z Wilson[1,3,5,6]*

[1]Interdisciplinary Program in Quantitative Biosciences, University of California Santa Barbara, Santa Barbara, United States; [2]Department of Bionanoscience, Kavli Institute of Nanoscience Delft, Technische Universiteit Delft, Delft, Netherlands; [3]Department of Molecular, Cellular, and Development Biology, University of California Santa Barbara, Santa Barbara, United States; [4]Integrated Biosciences, Inc., Redwood, United States; [5]Center for Bioengineering, University of California Santa Barbara, Santa Barbara, United States; [6]Neuroscience Research Institute, University of California Santa Barbara, Santa Barbara, United States

*For correspondence:
M.S.Bauer@tudelft.nl (MB);
mzw@ucsb.edu (MZW)

[†]These authors contributed equally to this work

## eLife Assessment

This **important** work combines theoretical analysis with precise experimental perturbation to demonstrate a previously unappreciated quantitative characteristic of the Wnt signaling pathway, which is anti-resonance, or a suppression of pathway output at intermediate activation frequencies. This effect is demonstrated experimentally with **compelling** evidence from optogenetic stimulation in multiple cell types, alongside modeling results that corroborate the phenomenon. While the demonstration of this phenomenon has yet to be extended to fully physiological situations, its clear existence within optogenetically stimulated systems shows that it is likely a significant factor that contributes to the behavior of this central signaling pathway.

**Abstract** Cells process dynamic signaling inputs to regulate fate decisions during development. While oscillations or waves in key developmental pathways, such as Wnt, have been widely observed, the principles governing how cells decode these signals remain unclear. By leveraging optogenetic control of the Wnt signaling pathway in both HEK293T cells and H9 human embryonic stem cells, we systematically map the relationship between signal frequency and downstream pathway activation. We find that cells exhibit a minimal response to Wnt at certain frequencies, a behavior we term anti-resonance. We developed both detailed biochemical and simplified hidden variable models that explain how anti-resonance emerges from the interplay between fast and slow pathway dynamics. Remarkably, we find that frequency directly influences cell fate decisions involved in human gastrulation; signals delivered at anti-resonant frequencies result in dramatically reduced mesoderm differentiation. Our work reveals a previously unknown mechanism of how cells decode dynamic signals and how anti-resonance may filter against spurious activation. These findings establish new insights into how cells decode dynamic signals with implications for tissue engineering, regenerative medicine, and cancer biology.

## Introduction

Cells and tissues do not merely respond to static cues but process dynamic signals that encode crucial information about cell fate decisions. These signaling dynamics have gained significant attention due to the widespread use of live-cell fluorescent reporters that enable their visualization throughout a

variety of developmental and cell fate transitions (*Regot et al., 2014*; *Yoney et al., 2018*; *Rahman and Haugh, 2023*). For example, oscillations in conserved signaling pathways determine cell fate decisions in organisms ranging from yeast to human (*Matsuda et al., 2020*; *Mitchell et al., 2015*; *Casani-Galdon and Garcia-Ojalvo, 2022*) and have been shown to selectively up-regulate specific genes (*Wilson et al., 2017*; *Batchelor et al., 2008*). In embryonic development, it has become increasingly clear that dynamic signals encode information through waves (*Martyn et al., 2019*; *Chhabra et al., 2019*; *Li and Elowitz, 2019*; *Deneke and Di Talia, 2018*), which are experienced by individual cells as periodic pulses or oscillations (*Levine et al., 2013*; *Albeck et al., 2013*; *Aulehla and Pourquié, 2008*; *El Azhar et al., 2024*). Thus, understanding how cells decode dynamic signals is fundamental to unlocking insights into tissue development and regeneration.

In engineering, the concepts of resonance and anti-resonance illustrate how certain dynamic signals can be amplified or suppressed based on input frequency (*Arora, 2014*). When a system receives input at its resonant frequency, the response is amplified; in contrast, at an anti-resonant frequency, the response is diminished. While these principles have proven useful in engineering, their application to biological information processing remains largely unexplored. Recent advances in synthetic biology and cellular engineering have enabled the manipulation of signaling pathways with unprecedented precision, providing the opportunity to apply such engineering frameworks to cell fate decisions. Notably, optogenetic tools allow for reversible, rapid, and spatially confined activation of signaling pathways, creating a platform to explore the temporal dimension of signaling in high resolution (*Möglich and Moffat, 2010*). By systematically investigating the cellular responses to dynamical inputs, we can uncover the 'design principles' underlying cell signaling and control and apply these insights to optimize tissue engineering, regenerative therapies, and gain deeper insights into development.

A particular pathway that has both displayed a variety of dynamical signaling patterns (*van Zon et al., 2024*) (pulses *Kroll et al., 2020*), oscillations (*Mengel et al., 2010*; *Sonnen et al., 2018*), and wave patterns (*Martyn et al., 2019*; *Hubaud and Pourquié, 2014*) and that is involved in almost every developmental and regenerative cell fate decision is the Wnt signaling pathway. In addition to Wnt signals directing proliferation and differentiation of various adult stem cell niches, Wnt is canonically known for its role in specifying the primitive streak and the mesoderm germ layer during gastrulation of all higher vertebrates. Indeed, the Wnt pathway has a unique topology that suggests non-trivial, non-monotonic responses to dynamic inputs. Negative feedback has been described to act at every level of this pathway. At the receptor level, FZD/LRP6 receptors are internalized and degraded upon Wnt binding on the timescale of hours (*Semënov et al., 2008*). In the cytoplasm, changes in inclusion of β-catenin (β-cat), the Wnt transcriptional effector, into Wnt processing biomolecular condensates (called the destruction complex/DC) occur on the timescale of 10 s of minutes (*Li et al., 2012*; *Lach et al., 2022*). Finally, DC scaffold proteins, such as Axin, are transcribed and feed back onto DC activity on the timescale of hours (*Lach et al., 2022*). These layered feedback mechanisms suggest that the Wnt pathway can process a rich tapestry of temporal signals, making it a key candidate for investigating how cells interpret dynamic signals to drive precise developmental outcomes.

The Wnt pathway has been modeled as a system of ordinary differential equations (ODEs) (*Lee et al., 2003*; *Giuraniuc et al., 2022*). Yet, these models involve many parameters (>20) and still fail to mechanistically capture the cell biology which plays a critical role in Wnt signal transduction. For example, Wnt signaling kinases and scaffold proteins form a phase-separated biomolecular condensate with a complex nucleation landscape which is not captured by simple ODEs that assume the components to be well-mixed (*Giuraniuc et al., 2022*). In contrast, abstracted models with a restricted number of 'effective' or 'hidden' variables are less reliant on the exact molecular wiring and instead focus on capturing the essential behaviors that emerge from underlying, often unobserved, interactions. This flexibility enables these abstracted models to generalize across different signaling pathways that more faithfully represent the input-output experimental framework enabled by the combination of optogenetic tools and live-cell reporters. This generality is particularly valuable in describing developmental signaling pathways, where their inherent versatility and complexity defy rigid, topologically fixed models.

Here, we combine optogenetic control of the Wnt pathway with mechanistic and abstracted modeling to understand how the set of possible input dynamics are processed into cell fate decisions. We engineered a clonal cell line that contained both reversible, optogenetic control of the Wnt

pathway as well as live-cell reporters that enabled precise, quantitative control and measurements of the Wnt signaling. By varying Wnt signal durations and monitoring responses, we explored Wnt target activity for simple optogenetic input patterns and developed an ODE model to fit their dynamics. The model predicts anti-resonant frequencies at which the response is suppressed. Experimental validation in HEK cells confirmed these predictions. We then built a minimal description of anti-resonance in the Wnt pathway that abstracts the biochemical interactions into a single effective or hidden variable; we use the phrase 'hidden variable' in analogy with hidden layers in neural networks. This model allows us to tune the properties of the anti-resonance using only two parameters. To demonstrate the generality and developmental relevance of our model, we engineered opto-Wnt into H9 human embryonic stem cells (hESCs). The mesoderm cell fate decision exhibited a stark dependence on stimulation dynamics, underscoring the developmental importance of anti-resonant frequencies. Overall, we present the first systematic optogenetic screen on developmental signaling dynamics in a mammalian stem cell line and pave the way to harnessing signaling dynamics for regenerative medicine and tissue engineering.

## Results

### A human cell line for all optical visualization and control of Wnt signaling dynamics

To enable simultaneous, precise control and real-time visualization of Wnt signaling, we engineered a clonal HEK293T cell line with optogenetic control over Wnt pathway dynamics by fusing Cry2 (*CRY2/CRY2, 2025*) to the LRP6 co-receptor (*Bugaj et al., 2015*; *Metcalfe et al., 2010*) (from here on referred to as the opto-Wnt tool). To track pathway outputs at multiple levels, we used a CRISPR knock-in strategy to endogenously tag β-catenin (β-cat) with a custom tdmRuby2 fluorescent protein, enabling real-time observation of transcription factor accumulation and degradation. Additionally, to monitor Wnt target gene transcription, we integrated an 8X-TOPFlash-tdIRFP (*Korinek et al., 1997*) reporter through lentiviral transduction. This engineered cell line, designated as the Wnt I/O (Input/Output), is the first clonal cell line to offer both temporal control and live visualization of Wnt signaling, providing

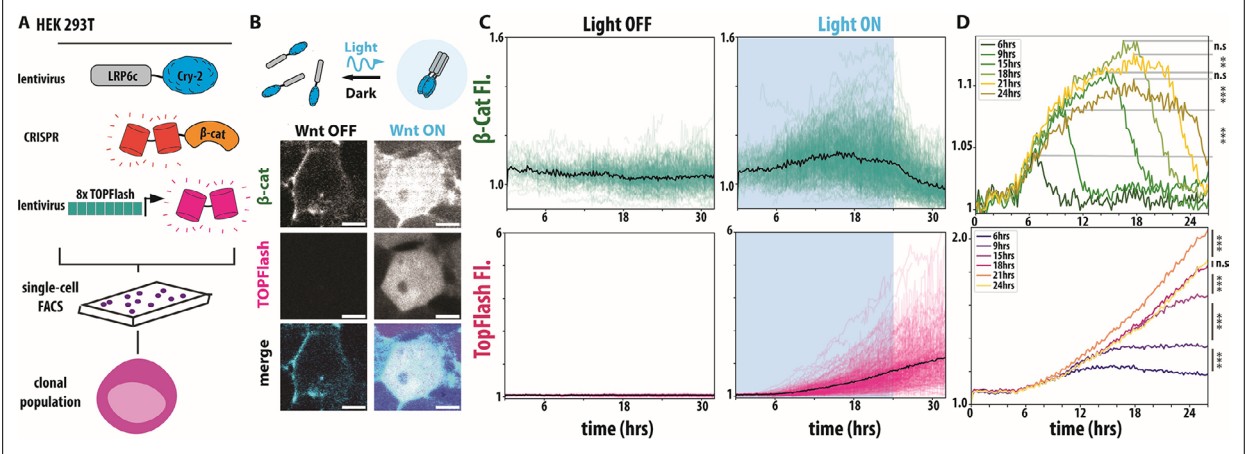

**Figure 1.** A human cell line for all optical visualization and control of Wnt signaling dynamics. (**A**) Schematic of HEK293T Wnt I/O cells containing lentiviral optogenetic LRP6c-Cry2Clust, CRISPR tdmRuby3-β-cat, lentiviral 8X-TOPFlash-tdIRFP, clonally FACS-sorted. (**B**) Live cell imaging of HEK293T Wnt I/O cells exposed to no light and 24 hr of 405 nm light illumination delivered every 2 min. Images are shown using the same lookup table. (**C**) Single-cell mean fluorescent intensity (MFI) traces (N=321–567 cells, four biological replicates per condition) of tdmRuby3-β-cat and (TopFlash) tdiRFP measurements from live HEK293T Wnt I/O cells tracked during exposure to activating blue light (right) or no light (left) controls. A blue background indicates light on, and a white indicates light off. Black line represents population mean. (**D**) Population means of live, single-cell β-catenin (top) and TopFlash (bottom) MFI traces from indicated conditions (N=321–595 cells, four biological replicates per condition, see Methods for significance values).

The online version of this article includes the following figure supplement(s) for figure 1:

**Figure supplement 1.** Optogenetic Activation of Wnt Signaling Enables Quantitative Single-Cell Analysis of β-Catenin Dynamics and Transcriptional Output.

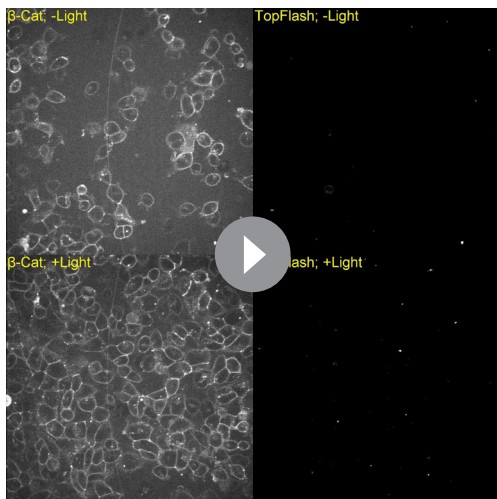

**Video 1.** Video of clonal HEK293T Wnt I/O under 24 hr of continuous light exposure, delivered as 100 ms long pulses every 2 min followed by an 8 hr relaxation period. *Top Left:* Wnt I/O HEK293T cell line β-catenin fluorescent channel, imaged every 10 min. In the absence of 405 nm blue light stimulation, β-catenin fails to accumulate in the nucleus due to no optogenetic activation of the opto-Wnt tool. *Top Right:* Wnt I/O HEK293T cell line TopFlash fluorescent channel, imaged every 10 min. In the absence of 405 nm blue light stimulation, TopFlash fails to accumulate due to no optogenetic activation of the opto-Wnt tool. *Bottom Left:* Wnt I/O HEK293T cell line β-catenin fluorescent channel, imaged every 10 minutes. In the presence of 405 nm blue light stimulation, β-catenin accumulates in the nucleus due to optogenetic activation of the opto-Wnt tool. *Bottom Right:* Wnt I/O HEK293T cell line TopFlash fluorescent channel, imaged every 10 min. In the presence of 405 nm blue light stimulation, TopFlash accumulates due to optogenetic activation of the opto-Wnt tool.

https://elifesciences.org/articles/107794/figures#video1

a platform for studying dynamic decoding in this pathway (*Figure 1A*). We confirmed that both reporters respond to 450 nm illumination through imaging and FACS (*Figure 1B*, *Figure 1—figure supplement 1A*).

Next, given the observed timescales of Wnt-induced cell fate decisions, we performed a baseline ON-OFF experiment to understand both the dynamics and heterogeneity of the Wnt I/O line. We tracked single-cell β-catenin and TopFlash dynamics in over 300 single cells during 24 hr of illumination to visualize pathway activation, followed by 8 hr of dark to visualize pathway de-activation (*Figure 1C*, *Video 1*). In comparison to the dark control, we noticed robust activation of the Wnt pathway on the population level, with variation in the activity patterns of individual cells.

Notably, TopFlash reporter activity did not decay in the dark, while β-catenin showed relatively fast degradation in the dark. Wide variation in Wnt response among cell populations has been widely reported in both optogenetic and non-optogenetic cell lines (*Figure 1—figure supplement 1B and C*, *Rahman and Haugh, 2023*; *Lach et al., 2022*; *Massey et al., 2019*). We attribute the observed cell population variation in both β-catenin and TopFlash to individual cells being in different phases of the cell cycle, which has been shown to influence Wnt signaling response (*Acebron et al., 2014*; *Davidson et al., 2009*). To quantify our videos, individual cells were segmented and tracked between each frame of our videos using a custom image analysis pipeline that involved the CellPose-Trackmate (*Stringer et al., 2021*) deep learning framework and focused on the nuclear fluorescence of β-catenin and TopFlash signals (*Figure 1—figure supplement 1D*, for details on tracking and segmentation see Methods). To reduce background noise in nuclear fluorescence signals, we first measured background fluorescence in the β-catenin and TopFlash channels for each frame. For each cell, raw nuclear β-catenin and TopFlash intensities were subtracted by the corresponding background value in that frame. The resulting traces were then normalized to the mean of all traces at t=0. As a control, we also normalized the mean fluorescence of β-catenin and TopFlash in the light-on condition to the light-off condition (*Figure 1—figure supplement 1E and F*).

In addition to long-duration Wnt signals (>24 hr), a wide range of shorter pulses of Wnt activity has been observed throughout mammalian development (*Mengel et al., 2010*; *Sonnen et al., 2018*). To

**Table 1.** P-value and corresponding symbol.

| p-value range | Symbol |
| --- | --- |
| $p>0.05$ | n.s. |
| $0.01<p\leq0.05$ | * |
| $0.001<p\leq0.01$ | ** |
| $p<0.001$ | *** |

understand the impact of these shorter Wnt pulses, we performed a Wnt duration scan, varying the activation time between 6 and 24 hr, followed by a variable rest time for a total 26 hr experiment. We found that each cell population could be distinguished by either its maximum β-catenin concentration or the final level of TopFlash with statistical significance and that our choice of normalization had no impact on our results. (*Figure 1D*, *Appendix 1—table 1*, *Figure 1—figure supplement 1G and H*). We also observe that the TopFlash signal is delayed from β-catenin, which responds earlier. Overall, these experiments demonstrate that our Wnt I/O line faithfully tracks and controls Wnt dynamics, allowing further interrogation of the signal processing capabilities.

Our quantifications reveal that β-catenin continues to accumulate in the presence of optogenetic stimulation, reaching a maximum at the end of optogenetic stimulation and decreasing in the dark. Our statistical analysis shows a significant difference in the 'light on' maximum fluorescent intensity for β-catenin between most conditions, with 18 and 24 hr showing no significance (*Figure 1D*, *Figure 1— figure supplement 1I*; significance thresholds defined in *Table 1*). This observation is consistent with β-catenin saturating to a maximum level during a prolonged Wnt activation, as described by previous experiments and models of the Wnt pathways (*Bugaj et al., 2013*). Looking at the TopFlash response, we see that longer light durations led to higher TopFlash levels at the end of the experiment and that post-stimulus TopFlash levels increase monotonically with light duration. The results of our statistical analysis revealed that all TopFlash conditions are also statistically significant compared to one another

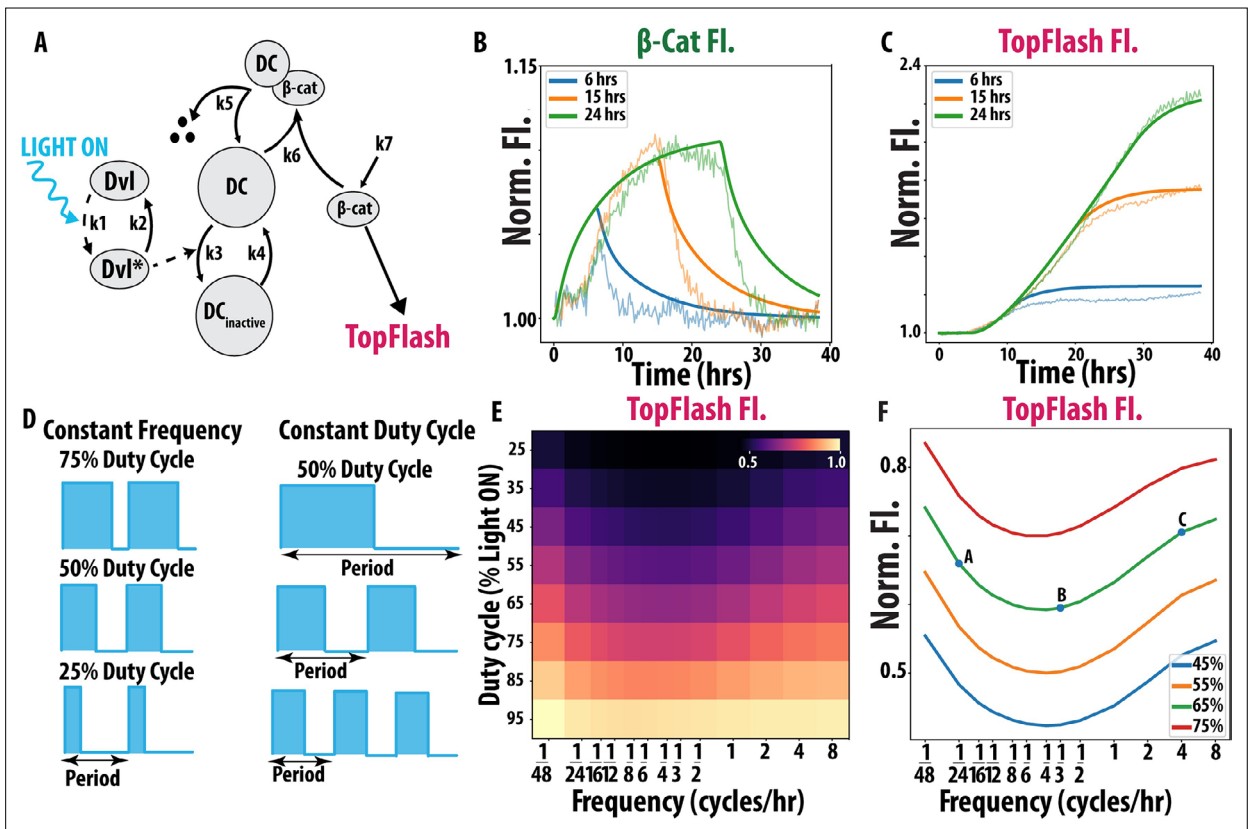

**Figure 2.** A model of Wnt signaling dynamics predicts anti-resonance. (**A**) Schematic of ordinary differential equation (ODE) model of Wnt signaling. Dotted lines represent light-dependent parameters. For information on model variables and parameters, refer to Methods, ODE Model for Wnt Signaling. (**B**) ODE model predictions (solid) of β-cat mean fluorescent intensity (MFI) for 6, 15, and 24 hr compared against our unsmoothed experimental results (light) from *Figure 1D*. Post-26 hr, experimental data was corrected for over-confluency effects in both β-catenin (β-cat) and TopFlash. (**C**) ODE model predictions (solid) of TopFlash MFI compared against our unsmoothed experimental results (light) from *Figure 1D*. (**D**) Visualization of duty cycle and frequency. *Left:* Constant frequency with varying duty cycle. *Right:* Constant duty cycle with varying frequency. (**E**) ODE model generated heatmap of endpoint TopFlash MFI for various combinations of duty cycle and frequency conditions. (**F**) Line graph of 45–75% duty cycles vs frequency with 1/24, 1/3, and 4 cycles/hr labeled as A, B, and C.

The online version of this article includes the following figure supplement(s) for figure 2:

**Figure supplement 1.** ODE-Based Predictions of Wnt Signal Decoding Are Recapitulated by Experimental TOPFlash Dynamics.

(*Figure 1D*, *Appendix 1—table 1*, *Figure 1—figure supplement 1J*). Overall, our data shows that our Wnt I/O cell line photo-switches on in light conditions and off in dark and can be used to quantify how temporal dynamics of Wnt signaling affect downstream gene expression.

## A model of Wnt signaling dynamics predicts anti-resonance

We then continued to analyze the ability of the Wnt signaling pathway to process dynamical inputs by developing a quantitative model. Although the Wnt pathway has been previously modeled, we sought to build a model of the Wnt pathway that captures our experimental observations with fewer parameters. We reduced the total number of differential equations in a previously established model of Wnt signaling (*Lee et al., 2003*; *Goentoro and Kirschner, 2009*; *de Man et al., 2021*) by describing only interactions between the DC and β-catenin (represented as state variables $c(t)$ and $b(t)$, respectively) (*Figure 2A*). The complex formed when β-catenin binds to the DC is represented by state variable $c_b(t)$. We represent optogenetic Wnt activation (denoted as $l(t)$) as directly increasing disassociation of the DC, omitting receptor-level dynamics of the pathway and instead modeling the optogenetic response through the activation and deactivation of Dvl (*Bugaj et al., 2015*) (represented as the state variable $d_a(t)$ when active and $d_i(t)$ when inactive). In total, we have nine $\{k_{1-7}, d_0, c_0\}$ parameters governing DC and β-catenin dynamics, with $k_{1-7}$ denoting rates and $d_0$ and $c_0$ denoting the conserved concentrations of Dvl and DC, respectively (Methods, Appendix 2). Since our experiments do not track all state variables directly, we use literature values (*Giuraniuc et al., 2022*; *Kang et al., 2022*; *Tan et al., 2012*; *Harris and Peifer, 2005*) to eliminate five parameters and constrain the remaining four parameters of the model using the experimental data (Methods and Appendix 2). We model TopFlash transcription (represented as the state variable $g(t)$) as a sigmoidal function of β-catenin accumulation. We also use experimental data to constrain the four free parameters $\{r_{max}, \tau, n, K\}$ governing TopFlash expression, whose expression activates non-linearly, with a Hill function with parameters $n$ and $K$, based on β-catenin levels at time $t - \tau$. We arrived at the following differential equations to describe Wnt signaling dynamics:

$$
\begin{aligned}
\frac{\mathrm{d}d_a}{\mathrm{d}t} &= k_1 l(t)\big(d_0 - d_a(t)\big) - k_2 d_a(t), \\
\frac{\mathrm{d}c}{\mathrm{d}t} &= -\big(k_3 d_a(t) + k_4 + k_6 b(t)\big)c(t) + k_4 c_0 + (k_5 - k_4)c_b(t), \\
\frac{\mathrm{d}c_b}{\mathrm{d}t} &= -k_5 c_b(t) + k_6 c(t)b(t), \\
\frac{\mathrm{d}b}{\mathrm{d}t} &= k_7 - k_6 c(t) b(t), \\
\frac{\mathrm{d}g}{\mathrm{d}t} &= r_{max} \frac{\big(b(t - \tau) - \bar{b}\big)^n}{\big(b(t - \tau) - \bar{b}\big)^n + K^n},
\end{aligned}
$$

where $\bar{b}$ denotes β-catenin at steady state when the light is off $l = 0$. We provide the complete list of parameters and their meaning in *Tables 2 and 3*.

We compare our model's β-catenin and TopFlash output for 6 hr, 15 hr, and 24 hr of continuous light exposure to the same experimental conditions in *Figure 1D*. We observe that our model recapitulates

**Table 2.** Model variables.

| Variable | Description |
|---|---|
| $d_a$ | Active disheveled |
| $d_i$ | Inactive disheveled |
| $c$ | Free destruction complex |
| $c_b$ | β-catenin bound to destruction complex |
| $c_i$ | Inactive destruction complex |
| $b$ | Free β-catenin |
| $g$ | TopFlash expression |
| $l$ | Light status ($l \in \{0, 1\}$) |

**Table 3.** Model parameters.

| Parameter | Dimensions* | Description |
|---|---|---|
| $k_1$ | min$^{-1}$ | Rate of disheveled activation |
| $k_2$ | min$^{-1}$ | Rate of disheveled deactivation |
| $k_3$ | min$^{-1}$ | Inactivation rate of destruction complex by disheveled |
| $k_4$ | min$^{-1}$ | Activation rate of destruction complex |
| $k_5$ | min$^{-1}$ | Dissociation of phosphorylated β-catenin from the destruction complex |
| $k_6$ | min$^{-1}$ | Binding and phosphorylation of β-catenin with/by destruction complex |
| $k_7$ | min$^{-1}$ | β-catenin synthesis rate |
| $d_0$ | . | Conserved concentration of disheveled |
| $c_0$ | . | Conserved concentration of destruction complex |
| $n$ | . | Hill coefficient |
| $K$ | . | Dissociation constant |
| $r_{max}$ | min$^{-1}$ | Maximum TopFlash transcription rate |
| $\tau$ | min | Time delay between β-catenin accumulation and onset of TopFlash transcription |

*In our system of units, concentration is dimensionless.

our experiment results (*Figure 1—figure supplement 1H and K*) (solid v. lighter line) (*Figure 2B and C*) with reasonable accuracy given cell-to-cell variability. This model, therefore, streamlines the complexity of Wnt signaling and still predicts β-catenin and TopFlash dynamics. While the mechanism proposed by the model has not been directly tested using experimental modification, such as specific inhibitors that perturb a single term in the model, our model agrees well with our experimental data. Assessing whether our model also reproduces the responses to these perturbations is an interesting direction for future work. We observe that our model matches TopFlash dynamics better than β-catenin dynamics. However, given the heterogeneity in expression profiles among the cell population and the use of literature values to constrain several parameters, our model still demonstrates a good agreement in both cases.

Using our model, we computationally explored the impact different complex temporal inputs have on Wnt signaling. First, we observe that TopFlash non-trivially accumulates in response to two Wnt signals of constant duration when the pause between them is varied (Appendix 2), reminiscent of non-trivial population dynamics outcomes under pulsed environmental variations (*Bauer et al., 2017*). This prompted us to investigate how Wnt signaling is affected when keeping the total duration of the Wnt signal constant while varying the number of pulses per unit time.

We screened through a large region of Wnt input space that represents observed signaling dynamics during development (*Martyn et al., 2019*; *Chhabra et al., 2019*). We systematically varied the total integrated light exposure (duty cycle) and the number of pulses per unit time (frequency) (*Figure 2D*). We simulated the β-catenin and TopFlash response of our Wnt model for 104 unique input combinations of duty cycle and frequency for 48 hr. We observe that β-catenin increases smoothly as frequency increases (*Figure 2—figure supplement 1A*) for a given duty cycle. Notably, for a given duty cycle, the total level of TopFlash fluorescence is lowest at an intermediate frequency: we observe a minimum at ~1/6 cycles/hr for all duty cycles (*Figure 2E*). Since TopFlash expression is reduced at these intermediate frequencies, we refer to this behavior as anti-resonance. Both the TopFlash heatmap as well as individual examples of constant duty cycle and varying frequency reveal a reduction in signal at intermediate frequencies consistent with anti-resonance (*Figure 2F*). Individual temporal traces of TopFlash from experiment and simulation at low, high, and anti-resonant frequencies (points A, B, C in *Figure 2F*) also demonstrated the anti-resonant effect (*Figure 2—figure supplement 21B and C*).

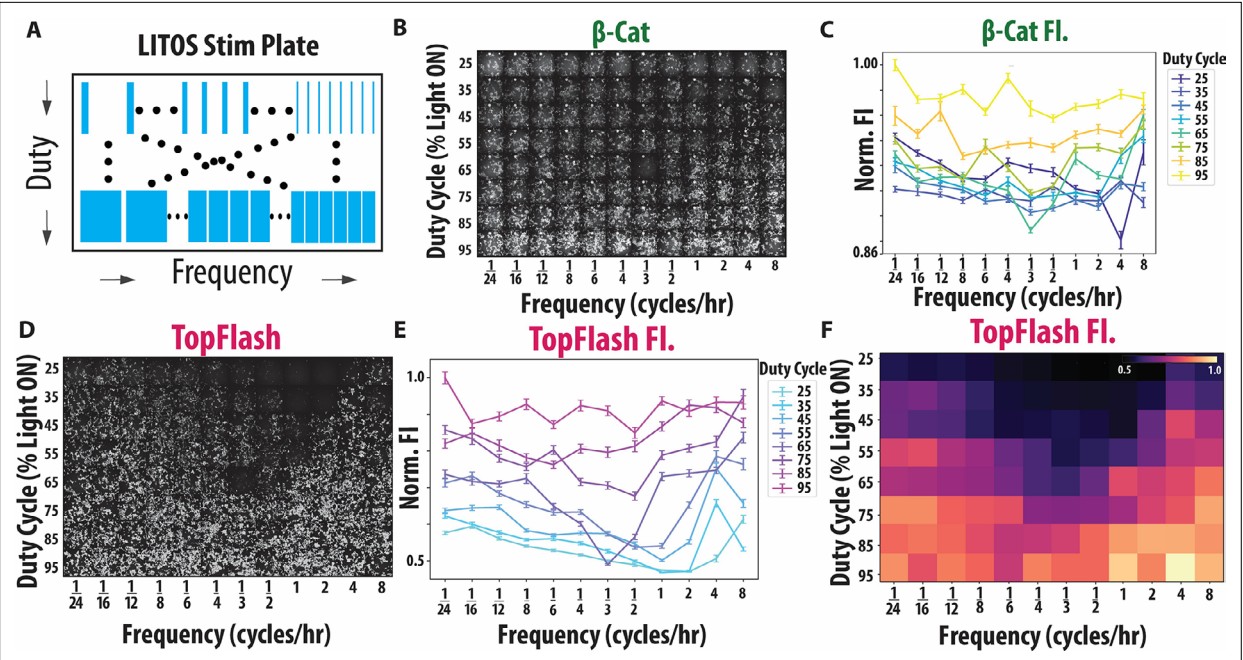

**Figure 3.** The Wnt pathway of HEK cells displays anti-resonance. (**A**) Schematic of our experimental method of the LITOS illumination device. (**B**) Qualitative images of end point β-catenin fluorescence post LITOS illumination with the heatmap of end point β-catenin mean fluorescent intensity (MFI) in the top right corner (N=106–590 cells, four biological replicates per condition). (**C**) Error bar plot of end point β-catenin MFI post frequency and duty cycle screen. Error bars represent standard error of the mean (SEM). (**D**) Qualitative images of end point TopFlash fluorescence post LITOS illumination (N=106–590 cells, four biological replicates per condition). (**E**) Error bar plot of end point TopFlash MFI post frequency and duty cycle screen. Error bars represent SEM. (**F**) Averaged heatmap of end point TopFlash MFI from two replicates of duty cycle and frequency experiment. Replicate heatmaps were normalized by the logarithm of the cell count at each well prior to averaging. Heatmap labels are displayed in categorical format, differentiating our experimental results heatmap from our computational heatmap.

The online version of this article includes the following figure supplement(s) for figure 3:

**Figure supplement 1.** Endpoint β-Catenin and Wnt Transcriptional Outputs Reveal Frequency-Dependent Signal Decoding.

## The Wnt pathway of HEK cells displays anti-resonance

Next, we tested if the anti-resonant frequencies predicted by the model occur in the Wnt I/O cell line. To simultaneously scan through a large range (=96) of unique duty cycle and frequency combinations, we utilized a high-throughput light stimulation device, the LITOS plate (*Höhener et al., 2022*), that can deliver unique light patterns to all individual wells of a 96-well plate (*Figure 3A*). Using this device, we performed the first-ever optogenetic screen of Wnt pathway dynamics over a 48 hr period.

We find that β-catenin is more uniformly expressed across frequencies than our model predicted, with some increase of β-catenin fluorescence for high frequency inputs (*Figure 2—figure supplement 1A*, *Figure 3—figure supplement 1A and B*). Based on our model, the β-catenin heatmap can be understood by the fast degradation times of β-cat (~10 min) relative to the timepoint of the last pulse of this experiment, meaning that β-catenin fluorescence after 48 hr depends also on the pulse arrangement in a particular frequency/duty cycle combination (*Figure 3B and C*, *Figure 3—figure supplement 1A and B*). In addition to quantifying fluorescence, we confirmed that there were no statistical differences due to cell number or total integrated light exposure (*Figure 3—figure supplement 1C and D*). Next, we look at TopFlash expression to verify the presence of the anti-resonance.

TopFlash displayed a clear anti-resonant effect with minimum activity between 1/3 and 1 cycles/hr, demonstrating that cells with the same duty cycle but different frequencies have varying pathway outputs (*Figure 3D*, *Figure 3—figure supplement 1E*). For example, the 55% duty cycle stimulation resulted in a clear decrease in TopFlash at the ~1/2 cycles/hr frequency (*Figure 3E*). We further verified the robustness of our results by performing a replicate experiment where we rearranged the order of the duty cycle and frequency conditions, demonstrating that the anti-resonance is independent of the experimental arrangement (*Figure 3—figure supplement 1F*). We quantified the TopFlash mean

fluorescent intensity (MFI) of our replicate experiment and averaged them with our original experiment's TopFlash MFI. Plotting the average of our two experiments as a heatmap, we observe a strong agreement between the results of our two experiments (*Figure 3F*, *Figure 3—figure supplement 1G and F*). These experimental results confirm anti-resonance in Wnt signaling and further confirm the predictions by our computational model of the Wnt pathway.

## Hidden-variable approach reveals timescales of Wnt activation define shape of anti-resonance

We have shown that a biochemical model based on established models of the canonical Wnt pathway correctly predicts the anti-resonance observed in optogenetic Wnt activation. Next, we abstract our model further to establish a minimal model that explains this non-monotonic behavior. Rather than

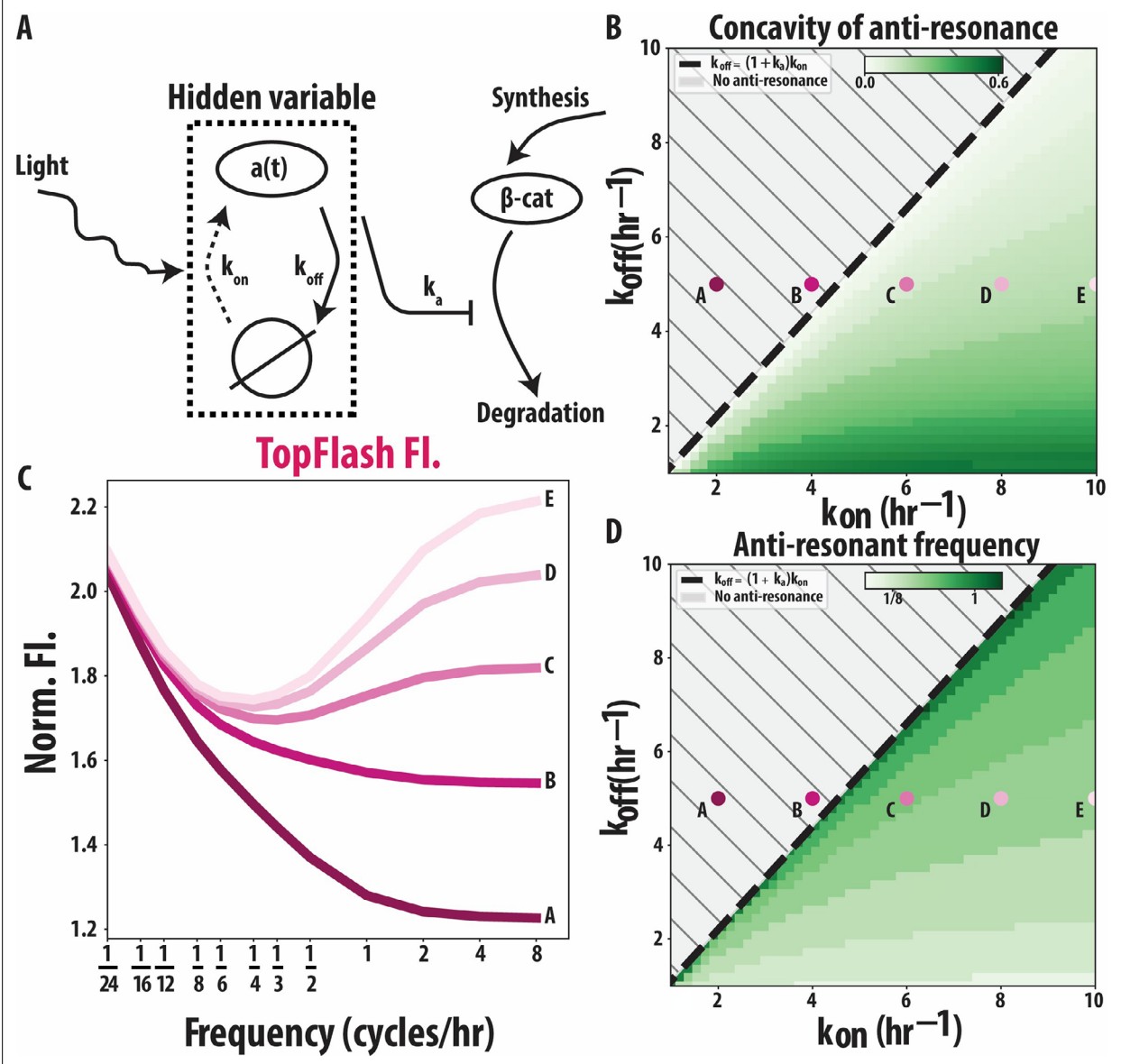

**Figure 4.** A hidden variable approach relates the anti-resonance to the timescales of Wnt activation and deactivation. (A) A hidden variable $a(t)$ is activated upon optogenetic Wnt activation at a rate $k_{on}$ and deactivates at rate $k_{off}$ when the light is turned off. In turn, $a(t)$ is coupled to first-order β-catenin dynamics. $b(t)$ (B–D) Systematic exploration of the parameter space shows that the rates $k_{on}$ and $k_{off}$ tune the concavity. (B) Concavity of anti-resonance is dependent on the combination of $k_{on}$ and $k_{off}$ rates. (C) Shape of the anti-resonance for five different points (A–E) in parameter space. As we enter the region $k_{off} < (1 + k_a) k_{on}$ the anti-resonance appears, consistent with our analytical result (see Appendix 3 for details about equations and parameter values). (D) Anti-resonant frequency is dependent on the combination of $k_{on}$ and $k_{off}$ rates.

explicitly modelling the interactions of proteins upstream of β-catenin, we abstract the time-dependent response of the Wnt pathway into a single 'hidden variable' $a(t)$ (*Figure 4A*). This variable is activated optogenetically until saturation at a rate $k_{on}$, and deactivated in the absence of light at a rate $k_{off}$. We use equivalent β-catenin dynamics to the ones earlier, with a degradation and a synthesis term, consistent with (*Goentoro and Kirschner, 2009*). The degradation of β-catenin dynamics is inhibited by the hidden variable, such that the β-catenin accumulates during optogenetic stimulation as before. We describe TopFlash transcription using the same Hill-type activation as the biochemical model in *Figure 2* and note that the coefficient of the Hill function has no significant effect on the presence of the anti-resonance.

If the dynamics of the hidden variable occur at shorter timescales than the dynamics of β-catenin, we can reproduce the dynamics of β-catenin and TopFlash from *Figure 2* as well as the anti-resonance (Appendix 3). Our more abstract model allows us to see that the anti-resonance arises because of the interplay of the timescales involved. In the low-frequency regime, the timescale of $a(t)$ becomes irrelevant, and only the slower β-catenin dynamics are important. When the frequency is increased, the pulse duration is no longer long enough for β-catenin to saturate to a steady-state value. Especially when coupled with non-linear Hill-type activation, this leads to less TopFlash being produced. As the frequency increases further towards the high-frequency regime, the dynamics of $a(t)$ become important. In this regime, we obtain increasing TopFlash expression with frequency as long as $k_{off} < (1 + k_2) k_{on}$ is satisfied. We confirm this bound analytically in Appendix 3. In *Figure 4B and D*, we quantify the concavity of the anti-resonance and the corresponding anti-resonant frequency, respectively. As $k_{on}$ increases with $k_{off}$ fixed, we observe that the minimum in the TopFlash expression becomes sharper, while the location of the minimum shifts to the left. For five different points on the $(k_{on}, k_{off})$-plane, we plot the final TopFlash level as a function of the frequency of the light cycles (*Figure 4C*). We observe in *Figure 4B and C* that the anti-resonant frequency indeed only appears once we enter the region $k_{off} < (1 + k_2) k_{on}$.

The explanation of anti-resonant behavior through two timescales in the hidden component(s) of the pathway suggests that the anti-resonance is a generic feature of the canonical Wnt pathway: it arises from differences in the timescale of activation and deactivation of the signaling cascade upstream of β-catenin. Mechanistically, the anti-resonant behavior is possible when the timescale of activation of the hidden variable is faster than its deactivation. In our full biochemical model, this can correspond, for example, to the phosphorylation timescale of Dvl protein at the receptor, which is thought to be fast for HEK293T cells (*González-Sancho et al., 2004*).

Our minimal model suggests that the anti-resonant suppression of regulatory responses at pulses of intermediate frequencies does not require fine-tuning to specific rates, but it does require these rates to satisfy a loose bound. Therefore, the generality of the model suggests that this behavior may also occur in other cells.

## Anti-resonant dynamics drive mesodermal stem cell differentiation in hESC H9s

To investigate whether Wnt pathway anti-resonance influences stem cell fate decisions in the human embryo, we explored its role in the differentiation of H9 human embryonic stem cells (hESCs). Indeed, Wnt signaling pulses, oscillations, and waves have all been observed in organoid models of human development (e.g. gastruloids, somatoids, embryoid bodies) (*Camacho-Aguilar et al., 2024*; *Shi, 2024*; *Aulicino et al., 2014*). Accordingly, we chose to address two questions: (1) Does anti-resonance affect the Wnt signals of human embryonic stem cells and, if so, (2) does it have a significant impact on developmental cell fate decisions?

We began by constructing a clonal H9 hESC line containing our optoWnt tool and CRISPR-tagged tdmRuby β-catenin (*Figure 5A*). First, we CRISPR-tagged the N-terminus of β-catenin in H9 hESCs with tdmRuby. We next used the piggyBac transposon system to introduce our opto-Wnt tool into these H9 cells, which were then selected for integrands and generated a clonal line (see Methods for details of cell line construction). Finally, we verified the opto-response by illuminating Wnt I/O H9s for 24 hr using 405 nm light and staining for Brachyury (BRA), a mesodermal cell fate marker (*Faial et al., 2015*). Optogenetic stimulation of our Wnt I/O H9 cell line resulted in efficient mesodermal differentiation (*Figure 5B*). From these observations, we conclude that our Wnt I/O H9 line responds to optogenetic activation and that this activation can control the cell fate decision of mesoderm differentiation.

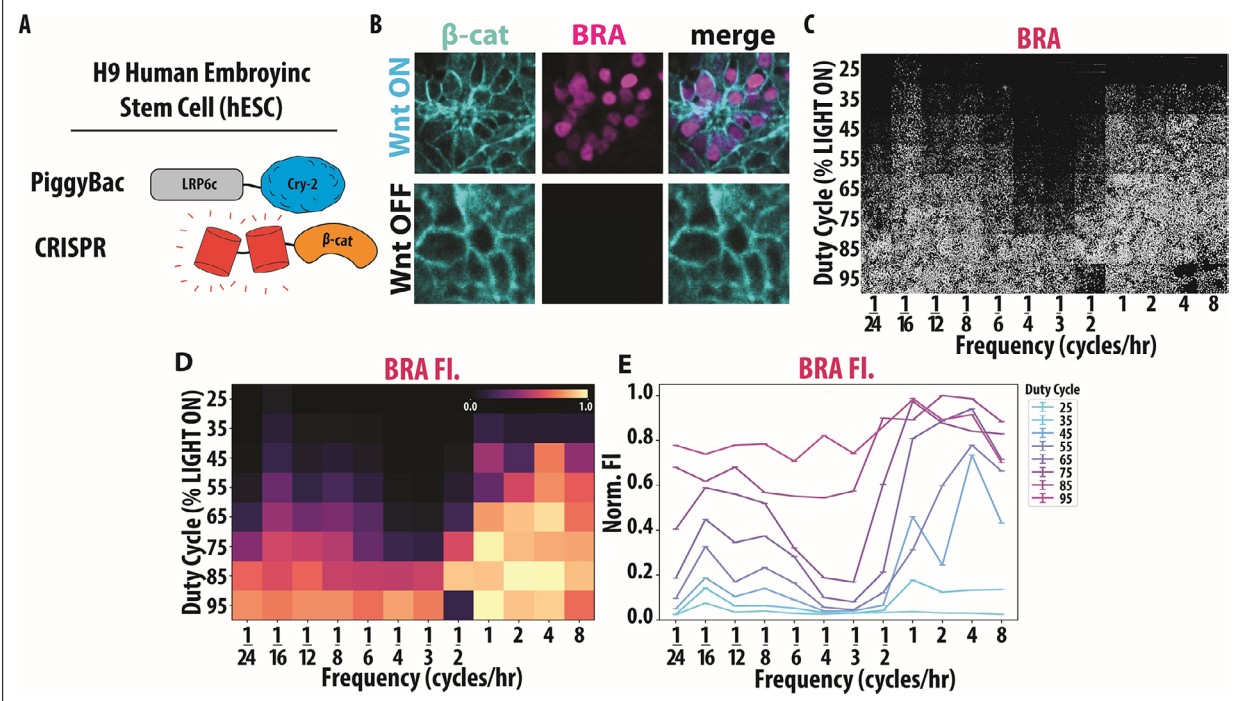

**Figure 5.** Anti-resonant dynamics drive mesodermal stem cell differentiation in human embryonic stem cell (hESC) H9s. (**A**) Schematic of H9 Wnt I/O cells containing PiggyBac optogenetic LRP6c-Cry2Clust and CRISPR tdmRuby3-β-catenin. (**B**) Representative examples of tdmRuby3-β-cat and Brachyury (BRA) accumulating in response to 24 hr of blue light activation in H9 Wnt I/O cells, post puromycin selection. (**C**) Qualitative images of the end point Brachyury (BRA) fluorescence post LITOS illumination (N=862–3176 cells, six biological replicates per condition). (**D**) Heatmap of end point BRA MFI for various duty cycle and frequency conditions. Heatmap labels are displayed in categorical format, differentiating our experimental results heatmap from our computational heatmap. (**E**) Error bar plot of end point BRA MFI post-frequency and duty cycle experiment. Error bars represent standard error of the mean (SEM).

The online version of this article includes the following figure supplement(s) for figure 5:

**Figure supplement 1.** Temporal Structure of Wnt Signaling Governs β-Catenin Dynamics and Mesoderm Marker Expression.

To determine whether anti-resonance affects mesoderm differentiation, we conducted a light stimulation screen targeting various duty cycles and frequencies, as described earlier (*Figure 3*). We focused on a 24 hr differentiation window (*Zhao et al., 2019*) as it is the literature-reported timescale for Wnt-driven mesoderm differentiation, and we found that using a 48 hr window caused all conditions to fully differentiate into mesodermal fates (*Figure 5—figure supplement 1A and B*). We applied the same stimulation parameters previously tested in the HEK293T line and following the stimulation period, cells were fixed and stained (*abcam, 2025*) for BRA. We then fluorescently imaged our cells for both β-catenin and BRA (*Figure 5C*, *Figure 5—figure supplement 1C, D and E*). Quantification of β-catenin agrees well with our model and HEK293T results (*Figure 5—figure supplement 1F*). Remarkably, quantification of BRA demonstrated a striking anti-resonant effect (*Figure 5D*). Namely, for cells with the same duty cycle (e.g. 55%), we observe that high and low frequencies induce total differentiation, while frequencies around 1/3 cycles per hour show markedly reduced BRA (*Figure 5E*). Together with our computational findings, these results demonstrate that anti-resonance is conserved across cell lines and that dynamic activation of the Wnt pathways influences stem cell differentiation in anti-resonant ways.

## Discussion

The concept that cells can interpret and respond to a diverse landscape of dynamic inputs has transformed our understanding of cellular signaling (*Wilson et al., 2017*; *Batchelor et al., 2008*; *Martyn et al., 2019*; *Chhabra et al., 2019*; *Li and Elowitz, 2019*; *Deneke and Di Talia, 2018*; *Levine et al., 2013*; *Albeck et al., 2013*; *Aulehla and Pourquié, 2008*; *El Azhar et al., 2024*). Here, we explored

this dynamic landscape by investigating the effects of periodic signals of varying frequency and duty cycle on the Wnt signaling pathway. Using a combination of computational, theoretical, and experimental approaches, we conducted the first optogenetic screen to probe the temporal dynamics of this essential signaling pathway in mammalian cells. We uncovered a previously unreported phenomenon in cellular signaling: for constant duty cycles, specific frequencies lead to significantly reduced levels of Wnt signaling. We called these anti-resonant frequencies, borrowing from engineering and electronics, where the term describes input frequencies that yield minimal system output. We confirmed the presence of the anti-resonance in both HEK293T cells and hESCs.

We speculate that the anti-resonance has biological meaning in that it suppresses a developmental response from oscillations with atypical timescales. For high-frequency oscillations, Wnt fluctuates rapidly, and it is likely not beneficial for the cells to respond to nuanced changes in fluctuations. For low frequencies, where the Wnt signal corresponds to pulses of multiple hours in duration, cells similarly require a strong and robust response. For intermediate frequencies, one might have expected a smooth interpolation between the low and high frequency regimes, but this does not occur; instead, the response to these intermediate frequencies is suppressed.

Analogous band-stop filtering should arise in other developmental circuits that couple a fast 'ON' step to slower deactivation or negative feedback. In Hedgehog, for example, PKA/CK1/GSK3-mediated partial proteolysis of Gli with slower recovery of full-length Gli creates the same fast-activation/slow-reset motif our hidden-variable model predicts will yield anti-resonance, and Wnt–Hedgehog crosstalk through the shared kinase GSK3 suggests such frequency selectivity could occur in other developmental signaling pathways (*Ding and Wang, 2017*). Especially in the context of Wnt, where individual receptors are internalized and degraded inside the cells for every binding ligand, requiring a detailed response to periodic inputs with fast periods would be cellularly costly (*Jiang et al., 2015*; *Yamamoto et al., 2006*). Different timescales for activation and deactivation likely also occur in different pathways, including the Erk (*Waters et al., 2014*; *Yadav et al., 2025*; *Ryu et al., 2015*), NF-kB (*Nelson et al., 2004*), BMP (*Teague et al., 2024*), Notch (*Weterings et al., 2024*), and Hippo/Yap pathways. For some of these pathways, input oscillations determine differentiation, often also in non-trivial ways depending on the period of the oscillations (*Yadav et al., 2025*; *Ryu et al., 2015*; *Weterings et al., 2024*). This suggests a non-trivial filtering of the input signals could be important in these pathways as well.

Anti-resonant frequencies could therefore represent a biological mechanism for noise filtering and signal discrimination. By suppressing specific input frequencies, cells can reduce spurious activation of downstream pathways, thereby improving the fidelity of information transmission. Here, we identify the anti-resonant frequency first in a Wnt reporter, whose expression is reduced at this frequency. When verifying its effect on the expression of the biologically meaningful differentiation reporter Bra, we observe a much stronger suppression at anti-resonance. In the context of oncogenesis, where intermediate Wnt is a hallmark of many cancers (*Zhao et al., 2022*; *Zhan et al., 2017*) (e.g. the 'Goldilocks Theory' *Albuquerque et al., 2002*), anti-resonance may act as a natural suppressor of intermediate signaling. This raises the intriguing possibility that the timescales and negative feedback inherent to Wnt signaling incorporate mechanisms to suppress even the most pernicious oncogenic signals. Indeed, it has been previously reported that certain frequencies of Wnt activation lead to human stem cell apoptosis (*Massey et al., 2019*).

Our observation is analogous to anti-resonant systems in an engineering context, where the amplitude of the response is suppressed at a particular anti-resonant frequency. We would like to highlight that at the moment, the observation of this phenomenon requires optogenetic screens, as the fast and sharp changes in signal concentrations are difficult to check using microfluidics. Furthermore, light delivery hardware allows for the testing of large sets of dynamical signals simultaneously, enabling the discovery of anti-resonant inputs.

Our results suggest that anti-resonance arises from the interplay of distinct timescales governing the synthesis, degradation, and interactions of components within the Wnt pathway. We found that a set of five ODEs, which simplify an existing model of the Wnt pathway, predicted the existence of an anti-resonant frequency, which we confirmed in experiments. To gain further insight, we coarse-grained the degrees of freedom that we do not observe in the experiments into a single hidden variable and showed that this abstracted model reliably captures the anti-resonant behavior. This reduction, inspired by observing sloppy parameters (*Gutenkunst et al., 2007*), preserves the model's

core predictive features, enhances its interpretability, and facilitates the extraction of meaningful insights. Here, we coarse-grained our model by considering the biochemical network and identifying the necessity for a hidden variable as a minimal effective model. Further work on how to consistently coarse-grain biochemical networks (*Xia et al., 2022*) to an input-output network with a minimum number of hidden or latent variables would provide a framework orthogonal and complementary to detailed modelling of molecular interactions. Here, this reduction allowed us to identify an interplay of timescales in the signaling pathway that makes possible the suppression of response to specific inputs. This increases our understanding of how such pathways 'compute' and what network architectures make these computations possible or efficient. We have shown that the optogenetic setup presents a toolbox to investigate hitherto unexplored input spaces.

## Outlook

Our work establishes an anti-resonant suppression of gene regulatory outputs for oscillatory signals at a frequency range. We connect this suppression to a hidden layer in the signaling network, which introduces an additional timescale to the problem. This consideration of different timescales is abundant in signaling cascades and the door to applying our approach to other signaling cascades with well-known dynamical responses, such as Erk (*Waters et al., 2014*; *Yadav et al., 2025*; *Ryu et al., 2015*), NF-kB (*Nelson et al., 2004*), BMP (*Teague et al., 2024*), and Notch (*van Oostrom et al., 2025*), which also exhibit oscillatory behavior and waves which can determine differentiation. Future work will explore whether anti-resonance is a universal feature of signaling networks and leverage this knowledge to uncover new regulatory mechanisms that may only become apparent in the context of highly dynamic signaling inputs.

# Materials and methods

## Cell lines

Human 293T cells were cultured at 37 °C and 5% $CO_2$ in Dulbecco's Modified Eagle Medium, high glucose GlutaMAX (Thermo Fisher Scientific, 10566016) medium supplemented with 10% fetal bovine serum (Atlas Biologicals, F-0500-D) and 1% penicillin-streptomycin. Experiments in human Embryonic Stem Cell (hESC) lines were performed using the H9 hESC cell line purchased from the William K. Bowes Center for Stem Cell Biology and Engineering at UCSB. Cells were grown in mTeSR Plus medium (Stem Cell Technologies) on Matrigel (Corning) coated tissue culture dishes and tested for mycoplasma in 2 month intervals.

## Wnt I/O 293T cell line generation

Clonal 293Ts containing CRISPR tdmRuby3-β-cat and oLRP6_Puro (AddGene ID: 249712) were obtained from Dr. Ryan Lach (*Lach et al., 2022*). Cells were co-transfected with pPig_8X-TOPFlash-tdIRFP_Puro (AddGene ID: 249713) obtained from Dr. Ryan Lach and Super PiggyBac Transposase (System Biosciences cat#: PB210PA-1) using manufacturers' recommendations and standard PEI-based transfection procedures. Cells were incubated for 24 hr before replacing them with fresh media. Cells were then subjected to 12 hr of continuous 405 nm light activation on the LITOS and single-cell FACS sorted for tdmRuby3+, tdIRFP + into a 96-well plate. Cells were monitored for growth over 14 days, only wells containing single colonies (arising from attachment of a single clone) were kept for subsequent processing. Prospective clonal populations were then imaged pre- and post-12 hr light activation to screen for low baseline expression of β-catenin and TopFlash and medium-high expression post-activation.

## Wnt I/O H9 cell line generation

Clonal β-catenin reporter lines were generated through CRISPR/Cas9-mediated homology-directed repair using analogous methods to HEK293T counterparts. Accutase-digested single hESCs were seeded onto Matrigel-coated 12-well plates and transfected with Lipofectamine Stem Transfection Reagent (Invitrogen, STEM00015) according to manufacturer recommendations. Once the cells grew to confluency, they were selected with 2 µg/mL puromycin in mTeSR Plus. Clonal populations were isolated through single cell sorting with the SH800 Sony Cell Sorter and expanded in mTeSR Plus

medium. The resulting population was screened and periodically treated with puromycin to ensure they mimicked canonical Wnt signaling upon optogenetic stimulation.

## Optogenetic stimulation

Spatial patterning of light during timelapse fluorescent imaging sessions was accomplished via purpose-built microscope-mounted LED-coupled digital micromirror devices (DMDs) triggered via Nikon NIS Elements software. Stimulation parameters (brightness levels, duration, pulse frequency) were optimized to minimize phototoxicity while maintaining continuous activation of Cry-2. For DMD-based stimulation on the microscope, the final settings for 'Light ON' were 25% LED power ($\lambda$=455 nm), 2 s duration pulses every 30 s. For experiments that did not require frequent confocal imaging, cells were stimulated via a benchtop LED array purpose-built for light delivery to cells in standard tissue culture plates (LITOS) (**Höhener et al., 2022**). The same light delivery parameters were used for LITOS-based stimulation as for microscope-mounted DMDs. Light was patterned to cover the entire surface of intended wells of plates used, rather than a single microscope imaging field.

## CHIR and Wnt3a stimulation

HEK293T cells were treated using 10 µM of CHIR99201 (Stem Cell Technologies, 72052) or 2.5 nM of Wnt3a ligand (R&D Systems, 5036-WN) added into Dulbecco's Modified Eagle Medium, high glucose GlutaMAX (Thermo Fisher Scientific, 10566016) medium supplemented with 10% fetal bovine serum (Atlas Biologicals, F-0500-D) and 1% penicillin-streptomycin. Cells were then imaged with DAPI in the same media 16 hrs post addition of CHIR99201 and Wnt3a ligand.

## Antibodies and immunofluorescence

Primary antibodies used to stain for Brachyury and Sox2 in H9s were α-Sox2 (Cell Signaling 3579, 1:500 dil.) and α-Brachyury (RnD AF2085, 1:500). Secondary antibodies used were α-Rbt-Alexa-488 (Thermo Fisher A21206, 1:1000) and α-Gt-Alexa-647 (Thermo Fisher A21447, 1:1000). Tissue fixation and staining was carried out using standard protocols using cold methanol (**abcam, 2025**). Immunofluorescent samples were imaged using confocal microscopy (see below). Nuclear stains were carried out using NucBlue Live ReadyProbes (Hoescht 33342, R37605) according to manufacturer's instructions.

## Imaging

All live and fixed cell imaging experiments were carried out using a Nikon W2 SoRa spinning-disk confocal microscope equipped with an incubation chamber maintaining cells at 37 °C and 5% CO2. Glass-bottom culture plates (Cellvis # P96-1.5H-N) were pre-treated with bovine fibronectin (Sigma #F1141) in the case of 293Ts or Matrigel in the case of H9s, and cells were allowed to adhere to the plate before subsequent treatment or imaging. HEK293T cells were imaged in Dulbecco's Modified Eagle Medium, high glucose GlutaMAX (Thermo Fisher Scientific, 10566016) medium supplemented with 10% fetal bovine serum (Atlas Biologicals, F-0500-D) and 1% penicillin-streptomycin. H9s were imaged in mTeSR Plus medium.

## Image analysis

All quantification of raw microscopy images was carried out using the same general workflow: background subtraction > classification > measurement > normalization > statistical comparison. When possible, subcellular segmentation of nuclear fluorescence was performed via context-trained deep learning-based Cellpose 2.0 algorithm derived from the 'nuclei' or 'cyto2' pretrained models prepacked with the current Cellpose software distribution available here: https://github.com/mouseland/cellpose (**Stringer et al., 2021**; **Stinger et al., 2025**). Single-cell tracking and raw measurements were performed with the 'LAP Tracker' function in the TrackMate plugin for ImageJ available here: https://imagej.net/plugins/trackmate/ (**Stringer et al., 2021**; **Ershov et al., 2022**). Tracks containing fewer than 50 contiguous frames (spurious or exited camera field of view) were omitted from subsequent analysis. Mean fluorescent intensity of regions of interest were measured and subsequently processed. Raw measurements were compiled, processed, and plotted via Python (ver. 3.9.13) scripts, available on Zenodo under Creative Commons Attribution 4.0 International: https://doi.org/10.5281/zenodo.17874586 (**Rosen, 2025**).

## Data normalization for cell trajectories

We begin by using the segmented and tracked cells produced by our Image Analysis method above. Each segmented and tracked cell contains information of its nuclear β-catenin and TopFlash at each timepoint during our timelapse experiment. In order to reduce background noise in nuclear fluorescence signals, we first measured background fluorescence in the β-catenin and TopFlash channels for each frame. For each cell, raw nuclear β-catenin and TopFlash intensities were subtracted by the corresponding background value in that frame. The resulting traces were then normalized to their values at t=0, such that all traces being at a normalized fluorescent intensity of 1.

## Data normalization for duty cycle and frequency experiments

We begin by using the segmented cells produced by our Image Analysis method. We do not calculate tracks here since duty cycle and frequency screen experiments only include end point image, so there is no temporal data. Each segmented and tracked cell contains information of its nuclear β-catenin and TopFlash. We measured background fluorescence in the β-catenin and TopFlash channels for each condition in the duty cycle and frequency experiment. Each segmented cell was background-subtracted. We then calculated the mean β-catenin and TopFlash fluorescence for each condition. Finally, we normalized each condition to the maximum mean β-catenin or TopFlash fluorescence, making the range of normalized fluorescent intensities between 0–1. For *Figure 3F*, background subtraction was performed on both the original and replicate experiments. Subsequently, the TopFlash fluorescence for each well was scaled by a factor of log(N), where N represents the cell count in that well. After applying this scaling, we calculated the mean between the original and replicate datasets to generate a combined dataset. Finally, the combined dataset was normalized to its maximum value before being visualized as a heatmap.

## ODE model for Wnt signaling

A simple model describing β-catenin accumulation in response to Wnt activation using a single differential equation, derived by *Goentoro and Kirschner, 2009*, did not capture the results here. Adapting a more detailed model, for example, (*Lee et al., 2003*) involving over 20 parameters. Thus, after parameter considerations detailed in Appendix 2, we arrived at the following differential equations describing β-catenin and TopFlash dynamics. Variables and parameters are defined in *Tables 2 and 3* and below. The dynamics of β-catenin for a given Wnt input is governed by:

$$
\begin{aligned}
\frac{\mathrm{d}d_a}{\mathrm{d}t} &= k_1 l\left(t\right)\left(d_0 - d_a\left(t\right)\right) - k_2 d_a\left(t\right), \\
\frac{\mathrm{d}c}{\mathrm{d}t} &= -\left(k_3 d_a\left(t\right) + k_4 + k_6 b\left(t\right)\right) c\left(t\right) + k_4 c_0 + \left(k_5 - k_4\right) c_b\left(t\right), \\
\frac{\mathrm{d}c_b}{\mathrm{d}t} &= -k_5 c_b\left(t\right) + k_6 c\left(t\right) b\left(t\right), \\
\frac{\mathrm{d}b}{\mathrm{d}t} &= k_7 - k_6 c\left(t\right) b\left(t\right).
\end{aligned}
\tag{1}
$$

In addition, we have the following conserved quantities:

$$
\begin{aligned}
c\left(t\right) + c_b\left(t\right) + c_i\left(t\right) &= c_0, \\
d_a\left(t\right) + d_i\left(t\right) &= d_0.
\end{aligned}
\tag{2}
$$

We model the dynamics of TopFlash using a simple Hill-type activation function:

$$
\frac{\mathrm{d}g}{\mathrm{d}t} = r_{max} \frac{\left(b\left(t - \tau\right) - \bar{b}\right)^n}{\left(b\left(t - \tau\right) - \bar{b}\right)^n + K^n},
\tag{3}
$$

where we have added a time delay $\tau$ to represent the time delay between β-cat accumulation and TopFlash transcription.

## Abstract model with hidden variable

We coarse-grain the model from *Figure 2*, described by *Equations 1-3*, to identify the core features of the model that give rise to the anti-resonance. The details of the signaling cascade upstream of β-catenin are coarse-grained into a single hidden variable $a\left(t\right)$. We denote the presence of the stimulus at

time $t$ with $l(t) \in \{0,1\}$, with $l(t) = 1$ during 'light on' and $l(t) = 0$ during 'light off.' The variable $a(t)$ is activated at rate $k_{\text{on}}$ when $l(t) = 1$, and deactivated at rate $k_{\text{off}}$ when $l(t) = 0$, i.e.,:

$$\frac{da}{dt} = \begin{cases} k_{\text{on}}\left(1 - a(t)\right) & \text{if } l(t) = 1, \\ -k_{\text{off}}a(t) & \text{if } l(t) = 0. \end{cases} \tag{4}$$

Note that we have normalized $a(t)$ such that the steady state when $l(t) = 1$ is $a(t) = 1$. Next, we couple the hidden variable to the β-catenin dynamics $b(t)$. We take it to be of the form derived by *Goentoro and Kirschner, 2009*:

$$\frac{db}{dt} = k_b \left(1 - \frac{b(t)}{1 + k_a a(t)}\right), \tag{5}$$

where $b(t)$ has been rescaled so that the steady state when $l(t) = 0$ is $b(t) = 1$. Finally, we describe the TopFlash expression $g(t)$ via the Hill-type activation function in *Equation 3* from the previous section (ODE model for Wnt Signaling).

The dynamical model described by *Equations 3, 4, and 5* are summarized by the schematic in *Figure 4* in the main text. We confirm that this abstract model also fits the single-pulse data and refer to Appendix 3 for more details. We find that parameters $k_b$ and $k_a$ dictate the coarse-grained β-catenin dynamics in *Equation 5*, while the parameters $k_{\text{on}}$ and $k_{\text{off}}$ govern the existence and shape of the anti-resonance. In the numerical simulations, we observe that the anti-resonance occurs in a parameter regime $k_{\text{off}} < \left(1 + k_a\right) k_{\text{on}}$. We derive this bound in Appendix 3 and sketch out the procedure here.

A periodic light input, like in our experiments, is determined by the duty cycle $\delta$ and frequency $f$, with period $T = 1/f$. For low frequencies $T \sim 1/k_b$, we can ignore the comparatively fast changes in $a(t)$ and replace it in *Equation 5* with $l(t)$:

$$\frac{db}{dt} \approx k_b \left(1 - \frac{b(t)}{1 + k_a l(t)}\right). \tag{6}$$

In this regime, the mean β-catenin level during the experiment strictly decreases as a function of frequency (see Appendix 3). Since TopFlash expression approximately tracks mean β-catenin levels, it also decreases with frequency in this regime. We find that the exact functional form of the TopFlash expression (*Equation 3*), and the Hill parameter influence the slope.

Next, we consider the high-frequency limit where $T \ll 1/k_b$. In this regime, the duration of each light pulse is short compared to the timescale in which significant changes occur in β-catenin levels. Hence, we can solve for small oscillations of β-catenin $\Delta b(t)$ around a constant value $\widetilde{b}$:

$$b(t) = \widetilde{b} + \Delta b(t), \Delta b(t) \ll \widetilde{b}. \tag{7}$$

For a TopFlash reduction at intermediate frequencies, TopFlash should increase as the frequency increases towards the high-frequency limit. As the TopFlash activation function in *Equation 3* is strictly increasing in $b(t)$, and $\Delta b(t)$ is small, it is sufficient to show that $\widetilde{b}$ increases as a function of frequency.

To first order in $\Delta b(t)$, the β-catenin dynamics in *Equation 5* can be written as:

$$\frac{d\Delta b(t)}{dt} = k_b \left(1 - \frac{\widetilde{b}}{1 + k_a a(t)}\right), \tag{8}$$

which we can solve for $\Delta b(t)$. Then, by imposing periodic boundary conditions, we can solve for the constant $\widetilde{b}$ (see Appendix 3 for details). For $\widetilde{b}$ to increase as the frequency increases towards the high-frequency limit, we require that:

$$\lim_{T \to 0} \widetilde{b} > \lim_{T \to \infty} \widetilde{b}. \tag{9}$$

Substituting the above limits (derived in Appendix 3) yields the following condition:

$$k_{\text{off}} < \left(1 + k_{\text{a}}\right) k_{\text{on}}, \tag{10}$$

in agreement with results from the numerical simulations. All modeling code is available on GitHub and Zenodo under Creative Commons Attribution 4.0 International: https://github.com/olivierwitteveen/wnt_antiresonance_model and https://doi.org/10.5281/zenodo.14834328 (*Witteveen, 2025*).

## Acknowledgements

We acknowledge helpful discussion with EF Wieschaus, J Rufo, A Maynard, A Bond, and MA Morrissey. MB acknowledges funding from the NWO Talent/VIDI program (NWO/VI.Vidi.223.169). MZW acknowledges funding support from the NIH NICHD R01 HD108803-04.

## Additional information

### Competing interests

Ryan S Lach: employee of Integrated Biosciences, Inc. Maxwell Z Wilson: Maxwell Z. Wilson is Co-founder and Chief Scientific Officer of Integrated Biosciences, a biotechnology company; this role is unrelated to the submitted work. The other authors declare that no competing interests exist.

### Funding

| Funder | Grant reference number | Author |
| --- | --- | --- |
| NIH NICHD | HD108803-04 | Naomi Baxter<br>Ryan S Lach<br>Maxwell Z Wilson |
| NWO Talent/VIDI Program | VI.Vidi.223.169 | Olivier Witteveen<br>Marianne Bauer |

The funders had no role in study design, data collection and interpretation, or the decision to submit the work for publication.

### Author contributions

Samuel J Rosen, Data curation, Software, Formal analysis, Validation, Investigation, Visualization, Methodology, Writing – original draft, Writing – review and editing; Olivier Witteveen, Software, Formal analysis, Methodology, Writing – original draft, Writing – review and editing; Naomi Baxter, Data curation, Investigation, Methodology; Ryan S Lach, Conceptualization, Data curation, Formal analysis, Investigation, Methodology; Erik Hopkins, Data curation, Investigation; Marianne Bauer, Conceptualization, Formal analysis, Funding acquisition, Investigation, Methodology, Writing – original draft, Project administration, Writing – review and editing; Maxwell Z Wilson, Conceptualization, Software, Supervision, Funding acquisition, Visualization, Writing – original draft, Project administration, Writing – review and editing

### Author ORCIDs

Samuel J Rosen ⓘ https://orcid.org/0000-0002-2981-7335
Olivier Witteveen ⓘ https://orcid.org/0009-0004-3049-4344
Marianne Bauer ⓘ https://orcid.org/0000-0002-1191-986X
Maxwell Z Wilson ⓘ https://orcid.org/0000-0003-0768-7004

Reviewer #1 (Public review): https://doi.org/10.7554/eLife.107794.3.sa1
Reviewer #2 (Public review): https://doi.org/10.7554/eLife.107794.3.sa2
Author response https://doi.org/10.7554/eLife.107794.3.sa3

## Additional files

### Supplementary files
MDAR checklist

### Data availability
All code for our computational Wnt modeling is available on GitHub and Zenodo under Creative Commons Attribution 4.0 international: https://github.com/olivierwitteveen/wnt_antiresonance_model and https://doi.org/10.5281/zenodo.14834328. All numerical data and code for plotting are available on Zenodo under Creative Commons Attribution 4.0 international: https://doi.org/10.5281/zenodo.17874586.

The following datasets were generated:

| Author(s) | Year | Dataset title | Dataset URL | Database and Identifier |
|---|---|---|---|---|
| Witteveen O | 2025 | olivierwitteveen/wnt_antiresonance_model: v1.0 Anti-resonance in the Wnt pathway | https://doi.org/10.5281/zenodo.14834328 | Zenodo, 10.5281/zenodo.14834328 |
| Rosen S | 2025 | Numerical Data and Plotting Code for Rosen et. al 2025 | https://doi.org/10.5281/zenodo.17874586 | Zenodo, 10.5281/zenodo.17874586 |

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

## Appendix 1

**Appendix 1—table 1.** T-test results from TopFlash (TF) and β-catenin (β-cat) traces from *Figure 1D*.

| Names | P-value TF | P-value β-cat | P-value TF stars | P-value β-cat stars |
|---|---|---|---|---|
| 6HR and 9HR | 8.65E-20 | 5.94E-05 | *** | *** |
| 6HR and 15HR | 2.19E-61 | 4.76E-12 | *** | *** |
| 6HR and 18HR | 4.56E-94 | 6.43E-19 | *** | *** |
| 6HR and 21HR | 1.11E-85 | 2.46E-12 | *** | *** |
| 6HR and 24HR | 9.14E-05 | 2.60E-12 | *** | *** |
| 9HR and 15HR | 9.05E-24 | 1.06E-05 | *** | *** |
| 9HR and 18HR | 6.39E-49 | 3.20E-14 | *** | *** |
| 9HR and 21HR | 2.88E-54 | 2.41E-08 | *** | *** |
| 9HR and 24HR | 7.52E-40 | 4.73E-06 | *** | *** |
| 15HR and 18HR | 1.93E-08 | 2.92E-06 | *** | *** |
| 15HR and 21HR | 6.50E-23 | 0.00795 | *** | ** |
| 15HR and 24HR | 1.12E-95 | 0.90173 | *** | n.s. |
| 18HR and 21HR | 4.16E-08 | 0.05985 | *** | n.s. |
| 18HR and 24HR | 1.86E-139 | 3.08E-07 | *** | *** |
| 21HR and 24HR | 2.06E-124 | 0.00334 | *** | ** |

## Appendix 2

## Model for β-catenin and TopFlash dynamics

The results from *Figure 1D* show the β-catenin (β-cat) and TopFlash dynamics in response to a single pulse of light. *Figure 3* explores more complex patterns of light pulses, revealing a non-monotonic relationship between the dynamics of LRP6 activation and the level of TopFlash expression. Our goal is to formulate a minimal biochemical model that captures the β-cat and TopFlash dynamics observed in *Figure 1* and correctly predicts the anti-resonance observed in *Figure 3*.

Quantitative measurements of parameters in the canonical Wnt pathway are not widely available (*Lee et al., 2003*, *de Man et al., 2021*, *Kang et al., 2022*). The first computational model for Wnt signaling was developed for Xenopus egg extracts by *Lee et al., 2003* and has been widely used to quantify Wnt pathway dynamics in other systems. This model relies on parameters detailing protein–protein interactions, protein synthesis/degradation, and phosphorylation/dephosphorylation, as well as concentrations of key signaling proteins. However, these protein abundances can vary significantly between Xenopus extracts and mammalian cell lines (*Kang et al., 2022*), and hence it is difficult to apply this Xenopus model directly to our cells. Thus, we will develop a simple model in the following. We emphasize that our objective here is not to precisely calibrate biophysical parameters for the canonical Wnt pathway in HEK293T cells, but rather to show that the observed dynamics can emerge from known interactions within the pathway.

Modeling of β-cat dynamics is challenging as the protein is distributed between cytoplasmic and nuclear pools. Further, β-cat is an important part of the cell-to-cell adhesion complex (*Harris and Peifer, 2005*). The mechanism through which Wnt signaling affects subcellular distribution of β-cat is not fully understood. For the purposes of simplifying the model, we do not explicitly model β-cat localization.

We first attempt a model that captures β-cat dynamics on the most coarse grained level using a single differential equation:

$$\frac{\mathrm{d}b}{\mathrm{d}t} = \alpha_1 - \frac{\alpha_2 b\left(t\right)}{1 + \alpha_3 l\left(t\right)}, \tag{11}$$

where $\{\alpha_i\}$ are rate constants, $b\left(t\right)$ denotes the β-cat concentration, and $l\left(t\right) \in \{0, 1\}$ is the light signal. The first and second terms on the right-hand side capture β-cat synthesis and DC-dependent degradation, respectively. The latter is turned 'off' by the Wnt signal. This physically intuitive equation was derived by *Goentoro and Kirschner, 2009* from the Xenopus model (*Lee et al., 2003*) using separation of timescales. We find that this simple model can capture β-cat dynamics of *Figure 1* reasonably well. However, we find that it does not capture well the anti-resonance in *Figure 3*; even when coupled to non-linear TopFlash activation functions, we observed only weak effects for highly specific parameter combinations. Hence, we decided to extend the model.

We add a layer of complexity to the model by considering destruction complex (DC) dynamics. We coarse-grain the DC formation cycle by treating it as an assembled object, where the light causes the DC to break down into an 'inactivated' form. Binding of β-cat with the DC leads to the release of phosphorylated β-cat and recycling of the DC, and phosphorylated β-cat is subsequently degraded. Finally, we model TopFlash expression using a Hill-type function of the β-cat concentration. We find that this model phenomenologically captures the dynamics in *Figure 1* and *Figure 3*. However, we noticed that the anti-resonance in *Figure 3* could be better described by including additional upstream protein dynamics. We will address this latter observation in Appendix 3. With this added step, stimulation by light causes activation of the disheveled (Dvl) protein instead of affecting the DC directly, as shown in *Figure 2A*.

Thus, we arrive at the following differential equations describing β-cat and TopFlash dynamics. We define the variables and parameters in *Table 2* shown in our Results section but note that we typically use '$k$' to refer to rates and '$b$,' '$c$,' and '$d$' to refer to β-catenin, DC, and Dvl concentrations, respectively.

The dynamics of β-cat for a given Wnt input is governed by:

$$\frac{dd_a}{dt} = k_1 l(t) (d_0 - d_a(t)) - k_2 d_a(t),$$
$$\frac{dc}{dt} = -(k_3 d_a(t) + k_4 + k_6 b(t)) c(t) + k_4 c_0 + (k_5 - k_4) c_b(t),$$
$$\frac{dc_b}{dt} = -k_5 c_b(t) + k_6 c(t) b(t),$$
$$\frac{db}{dt} = k_7 - k_6 c(t) b(t). \tag{12}$$

In addition, we have the following conserved quantities:

$$c(t) + c_b(t) + c_i(t) = c_0,$$
$$d_a(t) + d_i(t) = d_0. \tag{13}$$

Like in the main manuscript, $d_i$ and $c_i$ denote the inactive forms of Dvl and the DC, respectively, and $c_b$ denotes the β-catenin bound form of the DC. We model the dynamics of TopFlash using a simple Hill-type activation function:

$$\frac{dg}{dt} = r_{max} \frac{(b(t-\tau) - \bar{b})^n}{(b(t-\tau) - \bar{b})^n + K^n}, \tag{14}$$

where we have added a time delay $\tau$ to represent the time delay between β-cat accumulation and TopFlash transcription.

Our model, while being significantly more compact compared to the model for Xenopus (*Lee et al., 2003*), correctly predicts non-trivial dynamics of TopFlash and β-cat accumulation in *Figures 1 and 3*. It shares similarities with the model presented in *de Man et al., 2021*. However, contrary to the latter, we do not consider shuttling of β-cat between the cytoplasm and nucleus, or explicitly model interactions of β-cat with transcriptional co-activators. In our experiments, we did not quantify the dynamics of the DC directly. Hence, we use a combination of experimental data and values taken from literature to constrain the parameters of our minimal model.

We calculate steady-state solutions and use these to fix parameter values, as discussed in the sections below. Setting all time derivatives in *Equation 1* to 0, we can solve for the steady state of the model. Let us denote the steady-state concentration with a horizontal bar. Setting $l = 0$ or $l = 1$ yields the steady-state solutions for light 'off' and 'on,' respectively:

$$\bar{d_a} = \frac{k_1 l}{k_1 l + k_2} d_0,$$
$$\bar{c} = \left( c_0 - \frac{k_7}{k_5} \right) \frac{k_4}{\bar{d_a} k_3 + k_4},$$
$$\bar{c_b} = \frac{k_7}{k_5},$$
$$\bar{b} = \frac{k_7}{k_6} \frac{1}{\bar{c}}. \tag{15}$$

Here, we use a combination of our experimental data and values obtained from literature to constrain the parameters of the model.

We work in dimensionless units of concentration where $\bar{b} = 1$ when the light is off, $l = 0$. This allows us to write:

$$c_0 = k_7 \left( \frac{1}{k_5} + \frac{1}{k_6} \right). \tag{16}$$

Furthermore, we estimate from *Figure 1* that $\bar{b} \approx 1.1$ when $l = 1$. This implies:

$$k_3 \approx 0.10 \frac{k_4}{d_0} \left( 1 + \frac{k_2}{k_1} \right). \tag{17}$$

Next, we assume that the ratio $\bar{b}/\bar{c}$ when $l = 0$ is similar to *de Man et al., 2021*. This yields:

$$k_6 \approx \frac{91.0 \, \text{nM}}{82.4 \, \text{nM}} k_7 \approx 1.1 k_7. \tag{18}$$

We also assume that the ratio $\bar{b}/\bar{c}_b$ is similar to *de Man et al., 2021*. This yields:

$$k_5 \approx \frac{91.0\,\text{nM}}{62.5\,\text{nM}}k_7 \approx 1.5k_7. \tag{19}$$

Finally, we assume the β-cat synthesis rate reported in *Lee et al., 2003*. We convert to our dimensionless units of concentration by using $\bar{b} = 91\text{nM}$ from *de Man et al., 2021*. We obtain:

$$k_7 \approx \frac{0.423\,\text{nM}\,\text{min}^{-1}}{91\text{nM}} \approx 4.6 \times 10^{-3}\,\text{min}^{-1}. \tag{20}$$

We also choose to set $d_0 = 1$. For the remaining parameters $k_1$, $k_2$, and $k_4$, we run a numerical least-squares optimization using the data for single pulses. We find that a range of values of $k_1$, $k_2$ and $k_4$ capture the dynamics and use $k_1 = 0.17\text{min}^{-1}$, $k_2 = 0.17\text{min}^{-1}$ and $k_4 = 0.050\text{min}^{-1}$ for the rest of the manuscript. For the TopFlash dynamics, we use $n = 2$, $r_{\max} = 0.11\,\text{min}^{-1}$, $K = 1.0$, and $\tau = 4.0\,\text{hrs}$.

Our model matches the data well for both TopFlash and β-catenin, especially given the single-cell variability. We observe that the match in β-catenin could have been improved further if we had added an additional time delay in its transcription; since previous work did not incorporate this delay, we chose not to capture this delay here and instead add it for TopFlash expression.

Next, we explore the behavior of the model for two light pulses when we vary the pause duration. We expected a non-trivial response to pairs of pulses, as population dynamics model systems exposed to pulse sequences can show non-trivial responses; for example, when bacterial populations are exposed to antibiotics, two shorter antibiotic pulses can achieve the same response as a long one (*Bauer et al., 2017*). We find that for short pauses $\lesssim 1$ hour, the final level of TopFlash expression does not change significantly (*Appendix 2—figure 1*). For longer pauses, TopFlash expression decreases when the pause duration is increased. When a very long pause is present, the final TopFlash expression is about ~30% lower compared to when there is no pause. The observed effect could be of importance, as it means that the organism does not need to sustain Wnt activation continuously to obtain the same differentiated output.

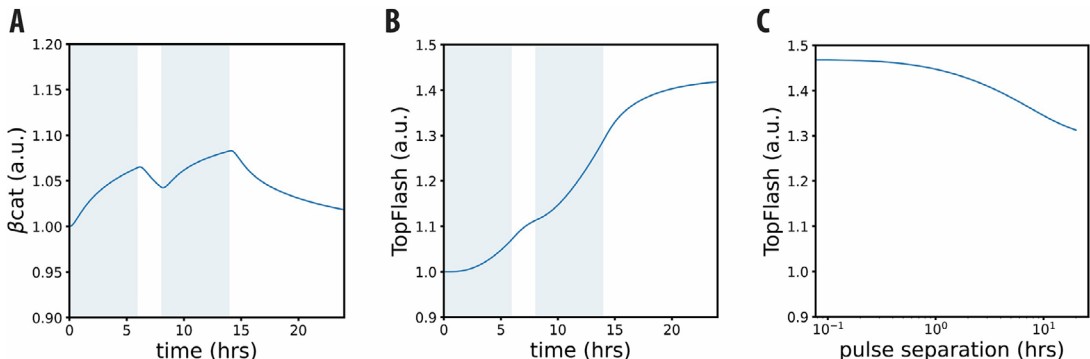

**Appendix 2—figure 1.** Inter-Pulse Timing Effects on β-Catenin and Wnt Transcriptional Memory. (**A**) β-catenin traces from computational model for two 6 hr pulses with a pause duration of 2 hr. (**B**) TopFlash traces from a computational model for two 6 hr pulses with a pause duration of 2 hr. (**C**) Final TopFlash expression level (at $t = 48$ hrs) after two 6 hr pulses with varying pause duration. Blue shading indicates opto-Wnt activation during these time intervals.

## Appendix 3

## Anti-resonance via a hidden variable

We observe an anti-resonance effect when the cells are subjected to periodic Wnt signals. That is, for the same amount of total integrated Wnt signal, intermediate frequencies result in the lowest amount of gene expression (*Figure 3*, *Figure 5*). We found that a simple biochemical model based on existing models of the canonical Wnt pathway, with parameter values from literature, correctly predicts the anti-resonance (*Figure 2*). By further coarse-graining the model, we aim to identify the core features of the model that give rise to the anti-resonance. In what follows, we seek to place constraints on model parameters for the anti-resonance effect to occur.

The canonical Wnt pathway has a time-dependent response – it takes time before changes in the light status propagate through the signaling cascade and affect β-cat levels. Intermediate steps include, for example, activation of disheveled protein and inactivation of the destruction complex. Rather than modelling these steps explicitly, we capture the time-dependent response of the pathway via a 'hidden variable' $a(t)$. Let us denote the light status with $l(t) \in \{0, 1\}$, where $l(t) = 0$ and $l(t) = 1$ correspond to 'light off' and 'light on' respectively. Let $a(t)$ be activated at rate $k_{on}$ when $l(t) = 1$, and deactivated at rate $k_{off}$ when $l(t) = 0$. The first-order dynamics of $a(t)$ are described by:

$$\frac{da}{dt} = \begin{cases} k_{on}(1 - a(t)) & \text{if } l(t) = 1, \\ -k_{off}a(t) & \text{if } l(t) = 0. \end{cases} \tag{21}$$

Note that we have normalized $a(t)$ such that the steady state when $l(t) = 1$ is $a(t) = 1$. Next, we couple the hidden variable to the β-cat dynamics $b(t)$. We take it to be of the form derived by *Goentoro and Kirschner, 2009*:

$$\frac{db}{dt} = k_b\left(1 - \frac{b(t)}{1 + k_a a(t)}\right), \tag{22}$$

where $b(t)$ has been rescaled so that the steady state when $l(t) = 0$ is $b(t) = 1$. Finally, we describe the TopFlash expression $g(t)$ via a Hill-type activation function like before:

$$\frac{dg}{dt} = r_{max}\frac{(b(t-\tau) - 1)^2}{(b(t-\tau) - 1)^2 - K^2}. \tag{23}$$

The dynamical model described by *Equations 11, 12, and 13* is summarized by the schematic in *Figure 4* in the main text.

For parameters $k_b$ and $k_a$ dictating the coarse-grained β-cat dynamics in *Equation 12*, we fit to the data from single-pulse experiments directly, as shown in *Appendix 3-figure 1A and B*. We find that $k_b = 3.3 \times 10^{-3}\,\text{min}^{-1}$ and $k_a = 0.10$ provide the best fit. Parameters for TopFlash transcription ($r_{max} = 0.11\,\text{min}^{-1}$, $K = 1.0$, and $\tau = 4.0\,\text{hrs}$) are kept the same as the model in *Figure 2*. We note that the parameters $k_{on}$ and $k_{off}$ dictating the dynamics of the hidden variable have no significant impact on the fit in *Appendix 3-figure 1A and B* as long as $k_{on}, k_{off} \gg k_b$. However, their values do govern the existence and shape of the anti-resonance. *Appendix 3-figure 1C* shows an example when $k_{on} = 8\,\text{hr}^{-1}$ and $k_{off} = 5\,\text{hr}^{-1}$.

We observe that the anti-resonance occurs in a parameter regime $k_{off} < (1 + k_a)k_{on}$. This is an interesting result, as it suggests that the anti-resonance is a generic feature of the canonical Wnt pathway: it arises due to the interplay between timescales in the problem. Crucially, it depends on the ratio between the timescales of activation and deactivation of the signaling cascade upstream of β-cat. We will now proceed to derive this bound analytically, and we show in the main manuscript that this bound matches the numerical simulations.

A periodic light input, like in the experiment, is determined by the duty cycle $\delta$ and frequency $f$, with period $T = 1/f$. The duration of each light pulse is given by $\delta \cdot T$. As such, we can write $l(t)$ as:

$$1(t) = \begin{cases} 1 & \text{if } 0 \leq \mathrm{mod}\,(t, T) < \delta T, \\ 0 & \text{if } \delta T \leq \mathrm{mod}\,(t, T) < T. \end{cases} \tag{24}$$

For this analysis, it is convenient to consider a quasi-steady state where the system has reached a periodic solution. Let us denote the solution for $a(t)$ in the 'light on' and 'light off' states as $a_{\mathrm{on}}(t)$ and $a_{\mathrm{off}}(t)$, respectively:

$$a(t) = \begin{cases} a_{\mathrm{on}}(t) & \text{if } 0 \leq \mathrm{mod}\,(t, T) < \delta T, \\ a_{\mathrm{off}}(t) & \text{if } \delta T \leq \mathrm{mod}\,(t, T) < T. \end{cases} \tag{25}$$

Writing the initial condition as $a(0) = a_0$, we can solve *Equation 11* to obtain:

$$\begin{aligned} a_{\mathrm{on}}(t) &= 1 - (1 - a_0)\, e^{-k_{\mathrm{on}} t}, \\ a_{\mathrm{off}}(t) &= a_{\mathrm{on}}(\delta T)\, e^{-k_{\mathrm{off}}(t - \delta T)}. \end{aligned} \tag{26}$$

We can solve for $a_0$ by demanding periodicity:

$$a_{\mathrm{on}}(0) = a_{\mathrm{off}}(T) = a_0, \tag{27}$$

which yields:

$$a_0 = \frac{e^{k_{\mathrm{on}} \delta T} - 1}{e^{(k_{\mathrm{on}} \delta + (1 - \delta) k_{\mathrm{off}}) T} - 1}. \tag{28}$$

First, let us consider the behavior of the model at low frequencies $T \sim 1/k_{\mathrm{b}}$. This represents the 'left side' of *Appendix 3-figure 1C*. In this regime, we can ignore the comparatively fast changes in $a(t)$ and replace it in *Equation 12* with $l(t)$:

$$\frac{db}{dt} \approx k_{\mathrm{b}} \left( 1 - \frac{b(t)}{1 + k_{\mathrm{a}} l(t)} \right). \tag{29}$$

Again, we consider the quasi-steady state and denote $b(t)$ piecewise as:

$$b(t) = \begin{cases} b_{\mathrm{on}}(t) & \text{if } 0 \leq \mathrm{mod}\,(t, T) < \delta T, \\ b_{\mathrm{off}}(t) & \text{if } \delta T \leq \mathrm{mod}\,(t, T) < T. \end{cases} \tag{30}$$

This *Equation 19* is separable, and we can solve for $b_{\mathrm{on}}(t)$ and $b_{\mathrm{off}}(t)$ directly by integrating. We obtain:

$$\begin{aligned} b_{\mathrm{on}}(t) &= (1 + k_{\mathrm{a}}) \left( 1 - e^{\frac{-k_{\mathrm{b}} t}{1 + k_{\mathrm{a}}}} \left( 1 - \frac{b_0}{1 + k_{\mathrm{a}}} \right) \right), \\ b_{\mathrm{off}}(t) &= 1 + (b_{\mathrm{on}}(\delta T) - 1)\, e^{-k_{\mathrm{b}}(t - \delta T)}, \end{aligned} \tag{31}$$

where we have denoted the initial condition $b(0) = b_0$. We can solve for $b_0$ by demanding that $b_0 = b_{\mathrm{off}}(0) = b_{\mathrm{on}}(T)$, which yields:

$$b_0 = 1 + \frac{k_{\mathrm{a}} \left( e^{\frac{k_{\mathrm{b}} \delta T}{1 + k_{\mathrm{a}}}} - 1 \right)}{e^{k_{\mathrm{b}}(1 - \delta) T + \frac{k_{\mathrm{b}} \delta T}{1 + k_{\mathrm{a}}}} - 1}. \tag{32}$$

Next, we can compute the mean β-cat during one cycle:

$$\frac{1}{T}\int_0^T b(t)\,dt = 1 + k_a\delta - \frac{k_a^2}{k_b T}\left(\frac{\left(1 - e^{-k_b(1-\delta)T}\right)\left(e^{\frac{k_b\delta T}{1+k_a}} - 1\right)}{e^{\frac{k_b\delta T}{1+k_a}} - e^{-k_b(1-\delta)T}}\right), \tag{33}$$

which is strictly decreasing as a function of frequency. This, intuitively, explains the decreasing TopFlash expression as the frequency increases in this regime. We note, however, that the functional behavior of TopFlash expression may not simply reflect the mean β-cat level during one period -- it depends on the specific TopFlash dynamics chosen. We find that a non-linear Hill-type activation (*Equation 13*) strengthens the downward slope, as periods of high β-cat achieved during long pulses are 'rewarded' with comparatively more TopFlash expression. Since in the low-frequency regime we have decreasing TopFlash expression as a function of frequency, an 'anti-resonant frequency' will appear only if TopFlash increases as we go to higher frequencies.

Next, we consider the high-frequency limit. More precisely, we consider the regime where $T \ll 1/k_b$. This is the 'right side' of *Appendix 3-figure 1C*. In this regime, the duration of each light pulse is short compared to the timescale in which significant changes occur in β-cat levels. Hence, we can solve for small oscillations of β-cat $\Delta b(t)$ around a constant value $\widetilde{b}$:

$$b(t) = \widetilde{b} + \Delta b(t), \Delta b(t) \ll \widetilde{b}. \tag{34}$$

To first order in $\Delta b(t)$, the β-cat dynamics in *Equation 12* can be written as:

$$\frac{d\Delta b(t)}{dt} = k_b\left(1 - \frac{\widetilde{b}}{1 + k_a a(t)}\right), \tag{35}$$

which is a separable equation. We can solve it directly by substituting our solution for $a(t)$ (*Equation 16*) and integrating. Denoting $\Delta b(t)$ piecewise as:

$$\Delta b(t) = \begin{cases} \Delta b_{on}(t) & \text{if } 0 \le \text{mod}(t,T) < \delta T, \\ \Delta b_{off}(t) & \text{if } \delta T \le \text{mod}(t,T) < T, \end{cases} \tag{36}$$

and imposing the initial condition $b(0) = \widetilde{b}$, we obtain:

$$\Delta b_{on}(t) = k_b\left[t - \frac{\widetilde{b}}{(1+k_a)k_{on}}\log\left(\frac{e^{k_{on}t}(1+k_a) - (1-a_0)k_a}{1 + a_0 k_b}\right)\right],$$
$$\Delta b_{off}(t) = \Delta b_{on}(\delta T) + k_b(t - \delta T) - k_b\left[\frac{\widetilde{b}}{k_{off}}\log\left(\frac{e^{k_{off}(t-\delta T)} + k_a a_{on}(\delta T)}{1 + k_a a_{on}(\delta t)}\right)\right]. \tag{37}$$

Like before, we can solve for $\widetilde{b}$ by demanding periodicity $\Delta b_{on}(0) = \Delta b_{off}(T) = \widetilde{b}$. This yields:

$$\widetilde{b} = (1+k_a)k_{off}k_{on}T\left[(1+k_a)k_{on}\log\left(\frac{e^{k_{off}(1-\delta)T} + a_{on}(\delta T)k_a}{1 + a_{on}(\delta T)k_a}\right)\right.$$
$$\left. + k_{off}\log\left(\frac{e^{k_{on}\delta T}(1+k_a) - (1-a_0)k_a}{1 + a_0 k_a}\right)\right]^{-1}. \tag{38}$$

Next, we consider the functional behavior of $a(t)$ with frequency. For anti-resonance to exist, we know that $\widetilde{b}$ must increase as the frequency increases towards the high-frequency limit. Formally,

one needs to show that (i) $\widetilde{b}$ is monotonic $T$ and (ii) determine the sign of $\mathrm{d}\widetilde{b}/\mathrm{d}T$. If $\mathrm{d}\widetilde{b}/\mathrm{d}T < 0$, anti-resonance occurs. Here, we take a shortcut by considering the limits of $T \to 0$ and $T \to \infty$. These are:

$$\lim_{T\to 0} \widetilde{b} = \frac{k_{\mathrm{off}}\left(1 - \delta\right) + \left(1 + k_{\mathrm{a}}\right) k_{\mathrm{on}}\delta}{k_{\mathrm{off}}\left(1 - \delta\right) + k_{\mathrm{on}}\delta}$$
$$\lim_{T\to\infty} \widetilde{b} = \frac{1 + k_{\mathrm{a}}}{1 + k_{\mathrm{a}}\left(1 - \delta\right)} \tag{39}$$

For anti-resonance, we require that:

$$\lim_{T\to 0} \widetilde{b} > \lim_{T\to\infty} \widetilde{b}. \tag{40}$$

Substituting the limits in *Equation 29* into the above inequality, we obtain the condition:

$$k_{\mathrm{off}} < \left(1 + k_{\mathrm{a}}\right) k_{\mathrm{on}}, \tag{41}$$

in agreement with results from the numerical simulations.

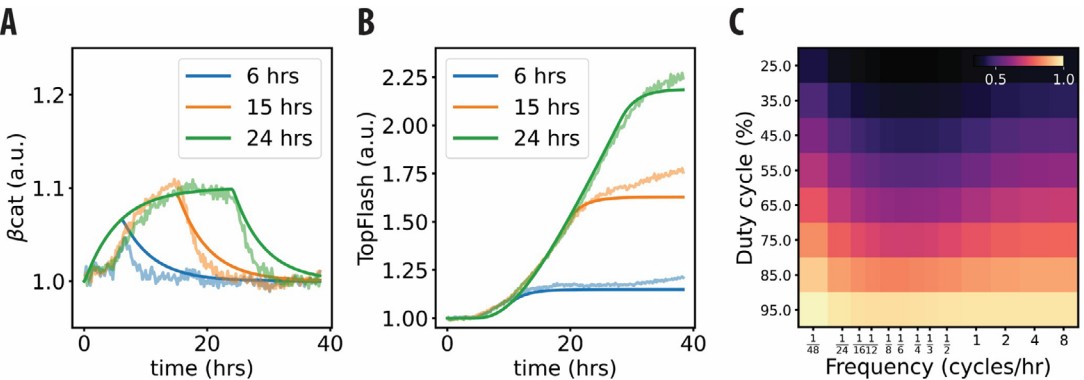

**Appendix 3—figure 1.** Hidden-Variable Model of β-Catenin and TOPFlash Dynamics at Multiple Light-Exposure Times. (**A**) Traces from the hidden-variable model of β-catenin at 6, 15, and 24 hr light exposure times. Dynamics in this regime are dictated by $k_{\mathrm{b}}$ and $k_{\mathrm{a}}$. Low-alpha lines show experimental data. (**B**) Traces from the hidden-variable model of TopFlash at 6, 15, and 24 hr light exposure times. Dynamics in this regime are dictated by $k_{\mathrm{b}}$ and $k_{\mathrm{a}}$. Low-alpha lines show experimental data. (**C**) TopFlash heatmap from the hidden-variable model displaying anti-resonance for $k_{\mathrm{on}} = 8\,\mathrm{hr}^{-1}$ and $k_{\mathrm{off}} = 5\,\mathrm{hr}^{-1}$.

