## [Editor Report · eLife Assessment]

This **important** work combines theoretical analysis with precise experimental perturbation to demonstrate a previously unappreciated quantitative characteristic of the Wnt signaling pathway, which is anti-resonance, or a suppression of pathway output at intermediate activation frequencies. This effect is demonstrated experimentally with **compelling** evidence from optogenetic stimulation in multiple cell types, alongside modeling results that corroborate the phenomenon. While the demonstration of this phenomenon has yet to be extended to fully physiological situations, its clear existence within optogenetically stimulated systems shows that it is likely a significant factor that contributes to the behavior of this central signaling pathway.

---

## [Referee Report · Reviewer #1 (Public review)]

Summary:

This report demonstrates that the gene expression output of the Wnt pathway, when controlled precisely by a synthetic light-based input, depends substantially on the frequency of stimulation. The particular frequency-dependent trend that is observed - anti-resonance, a suppression of target gene expression at intermediate frequencies given a constant duty cycle - is a novel aspect that has not been clearly shown before for this or other signaling pathways. The paper provides both clear experimental evidence of the phenomenon with engineered cellular systems and a model-based analysis of how the pairing of rate constants in pathway activation/deactivation could result in such a trend.

Strengths:

This report couples in vitro experimental data with an abstracted mathematical model. Both of these approaches appear to be technically sound and to provide consistent and strong support for the main conclusion. The experimental data are particularly clear, and the demonstration that Brachyury expression is subject to anti-resonance in ESCs is particularly compelling. The modeling approach is reasonably scaled for the system at the level of detail that is needed in this case, and the hidden variable analysis provides some insight into how the anti-resonance works.

In this revised manuscript, the authors have addressed issues in presentation and in discussing the broader relevance of their study to other pathways. Other limitations of the paper, including the fact that the anti-resonance phenomenon has not yet been demonstrated using physiological Wnt ligands and that the model has not been validated using experimental manipulations to establish that the mechanisms of the cell system and the model are the same, were deemed out of the scope of this initial demonstration by both the reviewers and authors. These questions will provide an interesting basis for further studies.

---

## [Referee Report · Reviewer #2 (Public review)]

Summary:

By combining optogenetics with theoretical modelling the authors identify an anti-resonance behavior in the WnT signaling pathway. This behavior is manifested as a minimal response at a certain stimulation frequency. Using an abstracted hidden variable model, the authors explain their findings by a competition of timescales. Furthermore, they experimentally show that this anti-resonance influences the cell fate decision involved in human gastrulation.

Strengths:

- This interdisciplinary study combines precise optogenetic manipulation with advanced modelling.

- The results are directly tested in two different systems: HEK293T cells and H9 human embryonic stem cells.

- The model is implemented based on previous literature and has two levels of detail: (i) a detailed biochemical model and (ii) an abstract model with a hidden parameter

Weaknesses:

- While the experiments provide both single-cell data and population data, the model only considers population data.

- Although the model captures the experimental data for TopFlash very well, the beta-Cat curves (Fig 2B) are only described qualitatively. This discrepancy is not discussed.

Overall Assessment:

The authors convincingly identified an anti-resonance behavior in a signaling pathway that is involved in cell fate decisions. The focus on a dynamic signal and the identification of such a behavior is important. I believe that the model approach of abstracting a complicated pathway with a hidden variable is an important tool to obtain an intuitive understanding of complicated dependencies in biology. Such a combination of precise ontogenetical manipulation with effective models will provide a new perspective on causal dependencies in signaling pathways and should not be limited only to the system that the authors study.

Comments on revisions:

I don't have any more comments for the authors and would like to congratulate them for the nice piece of work!

---

## [Author Response]

The following is the authors’ response to the original reviews

**Public Reviews:**

**Reviewer #1 (Public review):**
Summary:This report demonstrates that the gene expression output of the Wnt pathway, when controlled precisely by a synthetic light-based input, depends substantially on the frequency of stimulation. The particular frequency-dependent trend that is observed - anti-resonance, a suppression of target gene expression at intermediate frequencies given a constant duty cycle - is a novel aspect that has not been clearly shown before for this or other signaling pathways. The paper provides both clear experimental evidence of the phenomenon with engineered cellular systems and a model-based analysis of how the pairing of rate constants in pathway activation/deactivation could result in such a trend.Strengths:This report couples in vitro experimental data with an abstracted mathematical model. Both of these approaches appear to be technically sound and to provide consistent and strong support for the main conclusion. The experimental data are particularly clear, and the demonstration that Brachyury expression is subject to anti-resonance in ESCs is particularly compelling. The modeling approach is reasonably scaled for the system at the level of detail that is needed in this case, and the hidden variable analysis provides some insight into how the anti-resonance works.Weaknesses:(1) The anti-resonance phenomenon has not been demonstrated using physiological Wnt ligands; however, I view this as only a minor weakness for an initial report of the phenomenon. The potential significance of the phenomenon for Wnt outweighs the amount of effort it would take to carry the demonstration further - testing different frequencies/duty cycles at the level of ligand stimulus using microfluidics could get quite involved, and would likely take quite some time. Adding some more discussion about how the time scales of ligand-receptor binding could play into the reduced model would further ameliorate this issue.

We thank the reviewer for this comment and the interesting suggestion to test the anti-resonance phenomenon with microfluidics. We agree that combining physiological Wnt ligands with microfluidic stimulation would go beyond the scope of this current study, though it is an interesting extension. One advantage of the optogenetic setup, as mentioned in the discussion, is that the Wnt stimulus can be turned off sharply. This allows us to test the output from perfectly square wave input profiles; in microfluidics, washing the sticky ligand off the cells might “smear” the effective input profile cells respond to.

We show in Supplement Fig. 6, that our reduced model matches the experimental data and that we would expect the antiresonance phenomenon as long as \begin{document}$k_{\text {off }} \lesssim k_{\text {on }}$\end{document} (see Fig. 4). Practically, a smeared input profile implies an effective reduction of 𝑘_off_, which means that the phenomenon would be visible with microfluidics (provided the minimum is deep enough, see Fig. 4). However, this should still be considered with caution, as the antiresonance would then appear because the cells essentially receive a smeared out or continuous pulse in the high frequency limit, rather than cells responding to a square wave in a specific way.

(2) While the model is fully consistent with the data, it has not been validated using experimental manipulations to establish that the mechanisms of the cell system and the model are the same. There may be some ways to make such modifications, for example, using a proteasome inhibitor. An alternative would be to more explicitly mention the need to validate the model's mechanism with experiments.

We thank the reviewer for this valuable and constructive comment. We agree that future experimental perturbations that directly modulate pathway activation and reset kinetics—such as proteasome inhibition, targeted degradation of pathway components, or engineered changes in receptor turnover—would provide an important validation of the model’s mechanistic interpretation. In the present study, our primary goal was to establish the existence and quantitative features of anti-resonance in the Wnt pathway and to identify the minimal set of timescale relationships that can explain it. We view the proposed experimental validations as exciting next steps that extend beyond the scope of the current work, and we are grateful to the reviewer for emphasizing their importance. We now mention this explicitly in the discussion of our manuscript.

(3) I think the manuscript misses an opportunity to discuss the potential of the phenomenon in other pathways. The hedgehog pathway, for example, involves GSK3-mediated partial proteolysis of a transcription factor, which could conceivably be subject to similar behaviors, and there are certainly other examples as well.

We thank the reviewer for pointing out an opportunity to emphasize the possibility of this phenomenon in other pathways. The minimal model indicates that anti-resonance emerges whenever a rapid activating process is paired with a slower deactivating/reset process. Beyond Hedgehog/Gli processing, candidate circuits include: NF-κB (rapid IκBα phosphorylation/degradation vs slower IκBα resynthesis), ERK (fast phosphorylation bursts vs slower transcriptional negative feedback such as DUSPs), Notch (fast γ-secretase NICD release vs slower NICD turnover and feedback), BMP/TGF-β–SMAD (fast R-SMAD phosphorylation vs slower receptor trafficking/SMAD7 feedback), and Hippo/YAP (rapid cytoplasmic sequestration vs slower transcriptional feedback). Each contains the same timescale separation that should create a frequency ‘stop-band,’ predicting suppressed gene expression or fate transitions at intermediate stimulation frequencies. We have updated the manuscript’s discussion to mention the Hedgehog connection with the following added sentence in the discussion: Analogous band-stop filtering should arise in other developmental circuits that couple a fast ‘ON’ step to slower deactivation or negative feedback. In Hedgehog, for example, PKA/CK1/GSK3-mediated partial proteolysis of Gli with slower recovery of full-length Gli creates the same fast-activation/slow-reset motif our hidden-variable model predicts will yield anti-resonance, and Wnt–Hedgehog crosstalk through the shared kinase GSK3 suggests such frequency selectivity could occur in other developmental signaling pathways.

We also added an additional sentence regarding different activation and deactivation timescales in other pathways.

(4) Some aspects of the modeling and hidden variable analysis are not optimally presented in the main text, although when considered together with the Supplemental Data, there are no significant deficiencies.

We have addressed the model choices and analysis now more clearly in the main manuscript and also referred to the Supplemental Data more directly.

**Reviewer #2 (Public review):**
Summary:By combining optogenetics with theoretical modelling, the authors identify an anti-resonance behavior in the WnT signaling pathway. This behavior is manifested as a minimal response at a certain stimulation frequency. Using an abstracted hidden variable model, the authors explain their findings by a competition of timescales. Furthermore, they experimentally show that this anti-resonance influences the cell fate decision involved in human gastrulation.Strengths:(1) This interdisciplinary study combines precise optogenetic manipulation with advanced modelling.(2) The results are directly tested in two different systems: HEK293T cells and H9 human embryonic stem cells.(3) The model is implemented based on previous literature and has two levels of detail: (i) a detailed biochemical model and (ii) an abstract model with a hidden parameter.Weaknesses:(1) While the experiments provide both single-cell data and population data, the model only considers population data.

We thank the reviewer for correctly pointing out that the single-cell measurements would in principle allow us to incorporate the cell-to-cell heterogeneity into the model. In this study, we sought to identify a minimal quantitative model of the Wnt pathway that could explain anti-resonance through competing time scales. We believe that, for our purposes, focusing on population data allowed us to keep the complexity of the model to a minimum to increase its explanatory value. We agree with the reviewer that considering single-cell trajectories is an interesting direction for further work.

(2) Although the model captures the experimental data for TopFlash very well, the beta-Cat curves (Figure 2B) are only described qualitatively. This discrepancy is not discussed.

Indeed, our model fits to mean β-catenin expressions are more qualitative than for TopFlash. The fit for β-catenin was tricky, as expression of β-catenin is typically low and closer to the detectable limits than TopFlash. These experimental constraints mean that the variation between individual signal trajectories is higher for β-catenin compared to the light-off condition than for TopFlash. Therefore, we strove to obtain a qualitative rather than a quantitative fit to the mean expression profile in β-catenin. The current model fit is well within the standard deviation of variation. Given the observed heterogeneity and the fact that we take the parameters from literature (which ensures that the order of magnitude of parameters is in a sensible range), we believe that the model fits are reasonable. We now mention this explicitly in the text.

Overall Assessment:The authors convincingly identified an anti-resonance behavior in a signaling pathway that is involved in cell fate decisions. The focus on a dynamic signal and the identification of such a behavior is important. I believe that the model approach of abstracting a complicated pathway with a hidden variable is an important tool to obtain an intuitive understanding of complicated dependencies in biology. Such a combination of precise ontogenetic manipulation with effective models will provide a new perspective on causal dependencies in signaling pathways and should not be limited only to the system that the authors study.

We thank both reviewers for the positive assessment of our manuscript.

**Recommendations for the authors:**

**Reviewer #1 (Recommendations for the authors):**
There are several points that deserve more discussion, as noted above in the review.(1) It would be worthwhile to consider whether a relatively simple experiment with a proteasome inhibitor or similar pharmacological manipulation could provide useful validation data for the model.

We address this point above in the weaknesses section from reviewer 1.

(2) The figure legend for S5C should clarify whether the values plotted are at a particular fixed time point, or (more likely) at a certain time following the second pulse, which would be variable.

We have modified the figure caption to clarify that the values plotted are at a fixed time point in the simulation (*t*=48 hrs). We chose this timepoint sufficiently long after the second pulse to ensure that there are no residual dynamical effects. We thank the reviewer for noting this.

(3) As noted in the Sci Score document, various aspects of the resource reporter should be improved, such as including RRIDs, etc.

We are sending out our plasmids to AddGene; versions for Python and Matlab are listed in our methods section.

**Reviewer #2 (Recommendations for the authors):**
I mostly have suggestions to improve the clarity of the presentation.(1) Not all symbols in the equations given in the main text are explained. This is rather annoying, because either you present them and explain what they are or you don't show them and refer to the supplements. For example, \begin{document}$d_0$\end{document} or \begin{document}$c_o$\end{document} or \begin{document}$\bar{b}$\end{document} or n or K are not explained.

We have now more clearly presented the parameters in the main text and added signposts to the Methods section.

(2) Overall, it is often not clear what data in the figures are redundant, although the authors referred to them in the text. For example, in Figure 2c, a curve for 24 hours is shown and referred back to Figure 1D. However, in Figure 1D there is no curve for 24 hours. Is the data from Supplementary Figure 1 H and K also in the main text?

We thank the referee for pointing out these redundancies. We have now included the 24hr line in Figure 1D and are now only showing the unsmoothed data, also in the main text of the manuscript. To clarify supplemental figures, we have now removed S1H and S1K since all they showed was the unsmoothed version of the data. The remaining plots in Supplementary Figure 1 are normalized differently from what we show in Figure 1 to demonstrate our choice of normalization is not the reason for the observed optogenetic response.